

# Impact of post monsoon crop residue burning on PM$_{2.5}$ over North India: Optimizing emissions using a high-density in situ surface observation network

Mizuo Kajino[1,2,3], Kentaro Ishijima[1], Joseph Ching[4,3], Kazuyo Yamaji[5,3], Rio Ishikawa[6,1], Tomoki Kajikawa[6,1], Tanbir Singh[7], Tomoki Nakayama[8,3], Yutaka Matsumi[9,3], Koyo Kojima[6,1], Prabir K. Patra[3,10], and Sachiko Hayashida[3,11]

[1]Meteorological Research Institute (MRI), Japan Meteorological Agency (JMA), Tsukuba, Ibaraki 305-0052, Japan
[2]Faculty of Life and Environmental Sciences, University of Tsukuba, Tsukuba, Ibaraki 305-8572, Japan
[3]Research Institute for Humanity and Nature (RIHN), Kyoto, Kyoto 603-8047, Japan
[4]Department of Science and Environmental Studies, The Education University of Hong Kong, Hong Kong
[5]Graduate School of Maritime Sciences, Kobe University, Kobe, Hyogo 6580022, Japan
[6]Graduate School of Life and Environmental Sciences, University of Tsukuba, Tsukuba, Ibaraki 305-8572, Japan
[7]Shaheed Captain Vikram Batra Government College, Palampur, Himachal Pradesh 176061, India
[8]Faculty of Environmental Science, Nagasaki University, Nagasaki, Nagasaki 852-8521, Japan
[9]Institute for Space-Earth Environmental Research, Nagoya University, Nagoya, Aichi 464-8601, Japan
[10]Japan Agency for Marine-Earth Science and Technology (JAMSTEC) Yokohama, Kanagawa 236-0001, Japan
[11]Nara Women's University, Nara, Nara 630-8263, Japan

*Correspondence to*: M. Kajino (kajino@mri-jma.go.jp) and K. Ishijima (ishijima@mri-jma.go.jp)



**Abstract.** The impact of post monsoon crop residue burning (CRB) on surface $PM_{2.5}$ concentrations over the Punjab–Haryana–Delhi (PHD) region in North India was investigated using a regional meteorology–chemistry model, NHM-Chem, and a high-density in situ surface observation network comprising Compact and Useful $PM_{2.5}$ Instrument with Gas Sensors (CUPI-G) stations. We optimized CRB emissions from November 1 to 15, 2022 using NHM-Chem and surface $PM_{2.5}$

observational data. The CUPI-G data from Punjab was found to be crucial for CRB emission optimization, as the CRB emissions in North India in October and November are predominantly originating from Punjab, accounting for 80%. The new emission inventory is referred to as OFEv1.0, with 12 h time resolution, in daytime (5:30–17:30 IST) and nighttime (17:30–5:30 IST). The total emissions in OFEv1.0, such as $PM_{2.5}$, organic carbon, and black carbon, were consistent with previous studies, except CO, which was overestimated. OFEv1.0 substantially boosted emissions, which were

underestimated in satellite data due to clouds or thick haze on November 8 and 10, 2022. Large differences in optimized daytime and nighttime emissions indicated the importance of diurnal variations. Daytime emissions were larger than nighttime emissions on some days but not on others, indicating that diurnal variation shape may have differed each day. The mean contribution of CRB to surface $PM_{2.5}$ over PHD was 30%–34%, which increased to 50%–56% during plume events that transported pollutants from Punjab, to Haryana, to Delhi. Due to low performance of the meteorological simulation on

November 8 and 9, 2022, emission optimization was not successful in the case of increased $PM_{2.5}$ concentrations observed in Haryana on these days. The results of this study were obtained using a single transport model. Multi-model analysis is indispensable for better predictions and quantification of uncertainties in prediction results.

*Keywords:* NHM-Chem, Emission optimization, Low-cost sensors, Rice straw residue burning, Punjab, Source–receptor

relationship



## 1 Introduction

Delhi is a mega city with severe air pollution, and substantial efforts have been undertaken to understand the reasons underlying the rise in surface concentrations of air pollutants, source apportionments, impacts on human health, and mitigation policies and their effects (Rizwan et al., 2013; Guttikunda and Goel, 2013; Ghude et al., 2016; De Vito et al., 2018; Singh et al., 2019; Yadav et al., 2022; Lan et al., 2022; Guttikunda et al., 2023). Source apportionment studies indicate that vehicle exhaust, anthropogenic dust, biomass burning, and industry contribute approximately equally (10%–30%) to surface $PM_{2.5}$ concentrations in Delhi, although the dominant sector differs depending on the study (Yadav et al., 2022; Guttikunda et al., 2023). Delhi's air quality worsens during the post monsoon to winter period, which is associated with weaker wind speeds than other times of the year and increased emissions from space heaters (Guttikunda and Gurjar, 2012; Chowdhury et al., 2017, 2019; Guttikunda et al., 2023) and use of fireworks during Diwali festivities (Singh et al., 2019). In addition, Delhi's air quality is influenced by crop residue burning (CRB) of Kharif, which is grown in monsoon season and harvested in post monsoon season, in locations upwind of post monsoon northwesterly winds, such as Punjab and Haryana states (Cusworth et al., 2018; Beig et al., 2020; Takigawa et al., 2020; Liu et al., 2020; Singh et al., 2023). Indeed, post monsoon CRB emissions have increased since implementation of the groundwater conservation policy in 2009, which delays planting of Kharif crops and thus harvesting, leaving farmers with insufficient time to remove residues before planting Rabi crops (grown in winter), resulting in farmers burning their stubble (Balwinder-Singh et al., 2019; Mukherjee et al., 2023). However, a consensus has yet to be reached on the impact of post monsoon CRB emissions on regional air quality. For example, Cusworth et al. (2018) reported that CRB contributed 7.0%–78% of $PM_{2.5}$ primary components in Delhi depending on the year and selected emission inventory.

Source apportionment using 3D chemical transport models involves several uncertainties, the largest of which is derived from emission inventories. Constructing emission inventories can take a bottom–up or top–down approach. Bottom–up inventories are based on the amount of fuel (activity data) multiplied by the emissions of chemical compounds per unit mass of fuel (emission factor). Active fire data observed by polar-orbiting satellites are commonly employed as the activity data for bottom–up inventories of open biomass burning (e.g., Kaiser et al., 2012; Giglio et al., 2013; van der Werf et al., 2017; Beig et al., 2020; Liu et al., 2020; Singh et al., 2020; Wiedinmyer et al., 2023). Although substantial efforts have been undertaken to improve inventory data, large differences exist among emission inventories (e.g., Cusworth et al., 2018; Wiedinmyer et al., 2023). In the case of post monsoon CRB in North India, two main issues need to be resolved concerning emission inventories: (1) fire counts observed by satellites can be underestimated due to the presence of clouds or thick haze; (2) determining diurnal variations is impossible because polar-orbiting satellites travel once during the day, usually around noon (Takigawa et al., 2020; Liu et al., 2020). In fact, Takigawa et al. (2020) demonstrated that 3D dispersion simulations employing emission data based on MODerate resolution Imaging Spectroradiometer (MODIS) fire radiative power did not reproduce high $PM_{2.5}$ concentration episodes observed in Delhi in late October and early November 2019. To improve post monsoon CRB emission estimations in North India, Singh et al. (2020), Beig et al. (2020), and Liu et al. (2020) considered



small fires utilizing Visible Infrared Imaging Radiometer Suite (VIIRS) data, with a finer spatial resolution (375 m) than MODIS (1 km) data, which resulted in a 109% increase compared to using MODIS data alone in 2017 (Liu et al., 2020). Beig et al. (2020) combined geostationary satellite data with lower resolution (4 km) but higher temporal variation (15–30 min) to overcome the diurnal variation issue. Nevertheless, nighttime observational data are unavailable even though some
farmers ignite fires after sunset (Liu et al., 2020). Liu et al. (2020) tried to overcome both issues by (1) assuming a Gaussian distribution for spatial and temporal variations of CRB emissions and (2) employing a household survey of >2000 farmers.

Another useful way to improve emission estimations is through a top–down approach, which minimizes the cost function between simulations and observations by adjusting for emission fluxes (e.g., Yumimoto and Takemura, 2015). In many cases, satellite data are employed to constrain emission fluxes due to their large spatial coverage (Elguindi et al., 2020).
Surface observational data, which include more chemical compounds and have higher temporal variations than satellite data, can be utilized for limited regions (i.e., only over land because surface observations are scarce over oceans) if the spatial coverage is sufficiently high. For example, Henze et al. (2009) optimized inorganic $PM_{2.5}$ precursor emissions over the United States using Interagency Monitoring of PROtected Visual Environment (IMPROVE) monitoring datasets. However, to date, no studies have developed top–down inventories of post monsoon CRB emissions in North India.

India hosts a nationwide network of surface air quality monitoring stations, named Continuous Ambient Air Quality Monitoring (CAAQM) stations, and data are provided by the Central Pollution Control Board (CPCB). Because CAAQM stations are mostly located in large cities, surface $PM_{2.5}$ concentration data in CRB source areas have not been available. To address this issue, Singh et al. (2023) distributed low-cost sensors for air quality monitoring, known as Compact and Useful $PM_{2.5}$ Instrument with Gas Sensors (CUPI-G), over rural and farmland areas in the Punjab–Haryana–Delhi (PHD) region.
Together with meteorological analysis datasets, these authors successfully identified two transport events of air pollutants from Punjab, to Haryana, to Delhi in 2022.

Based on the work by Singh et al. (2023), the current study aimed to quantify the impact of CRB emissions on surface $PM_{2.5}$ concentrations (i.e., the so-called source–receptor relationship of $PM_{2.5}$) over the PHD region using a 3D regional meteorology–chemistry model (Kajino et al., 2019). Additionally, we applied an emission optimization technique to
develop a top–down CRB emission inventory by resolving underestimations due to clouds and thick haze and considering diurnal variations (12 h resolution, differentiating daytime and nighttime) using CUPI-G station data (Singh et al., 2023). To the best of our knowledge, this is the first study to apply a top–down approach to constrain post monsoon CRB emissions in North India. The optimized emissions were compared against other bottom–up inventories.



## 2 Methods

### 2.1 Numerical models and simulation settings

The Japan Meteorological Agency (JMA)'s regional meteorology–chemistry model, named Non-Hydrostatic Model (NHM)-Chem v1.0 (Kajino et al., 2019; 2021), was utilized herein. NHM-Chem v1.0 is a chemical transport model (CTM) coupled offline with NHM, the previous version of JMA's numerical weather prediction (NWP) model (Saito et al., 2006; 2007). The CTM part of offline-coupled NHM-Chem can be employed with other meteorological models, such as the current version of JMA's NWP model, A System based on a Unified Concept for Atmosphere (ASUCA; JMA 2014; Ishida et al., 2022), named ASUCA-Chem (Kajino et al., 2022), and Scalable Computing for Advanced Library and Environment (SCALE; Nishizawa et al., 2015; Sato et al., 2015), named Offline SCALE-Chem (Nakata et al., 2021; Sato et al., 2023a; 2023b). The CTM part of NHM-Chem is also utilized with the Weather Research and Forecast model (WRF; Skamarock et al., 2019). Offline coupling between WRF v4.1.5 and the CTM part of NHM-Chem v1.0 was employed in this study.

Fig. 1 presents the model domains utilized in the simulation. The mother domain (domain 1; D01) covered all of India, with a horizontal grid resolution of $\Delta x$ = 30 km to resolve the transport phenomena associated with synoptic scale circulations. The nested domain (domain 2; D02) covered the northwestern part of India, with $\Delta x$ = 6 km to resolve local transport phenomena such as mountain–valley circulation and mesoscale cloud processes. There were 32 vertical levels from the surface to 50 hPa for WRF and 40 vertical levels from the surface to 18 km for CTM using terrain-following coordinates. Two-way nesting was applied for WRF, with the initial and boundary conditions provided by the National Center for Environmental Prediction (NCEP) final operational global analysis data (available from https://rda.ucar.edu/datasets/ds083.2, last accessed: May 7, 2024) (1° × 1°, 6 hourly). The NCEP final analysis data were also utilized for grid nudging over D01. The climatological chemistry data provided by the Meteorological Research Institute Chemistry-Climate Model version 2 (MRI-CCM2; Deushi and Shibata, 2011) (TL159, $\Delta x$ ~ 1.125°, monthly) and Model of Aerosol Species in the Global Atmosphere mark-2 (MASINGAR mk-2; Tanaka et al., 2003; Tanaka and Ogi, 2017; Yumimoto et al., 2017) (TL159, $\Delta x$ ~ 1.125°, monthly) were employed for the initial and boundary concentrations of gaseous and aerosol species over D01, respectively. The boundary concentrations of gaseous and aerosol species over D02 were provided by the hourly simulation results of D01. The simulation period was from October 10, 2022 (00:00 UTC) to November 15, 2022 (00:00 UTC), with a 5-day spin-up period, resulting in an analysis period of 1 month from October 15, 2022 (00:00 UTC) to November 15, 2022 (00:00 UTC). The time interval of WRF output and CTM input/output was 1 h.



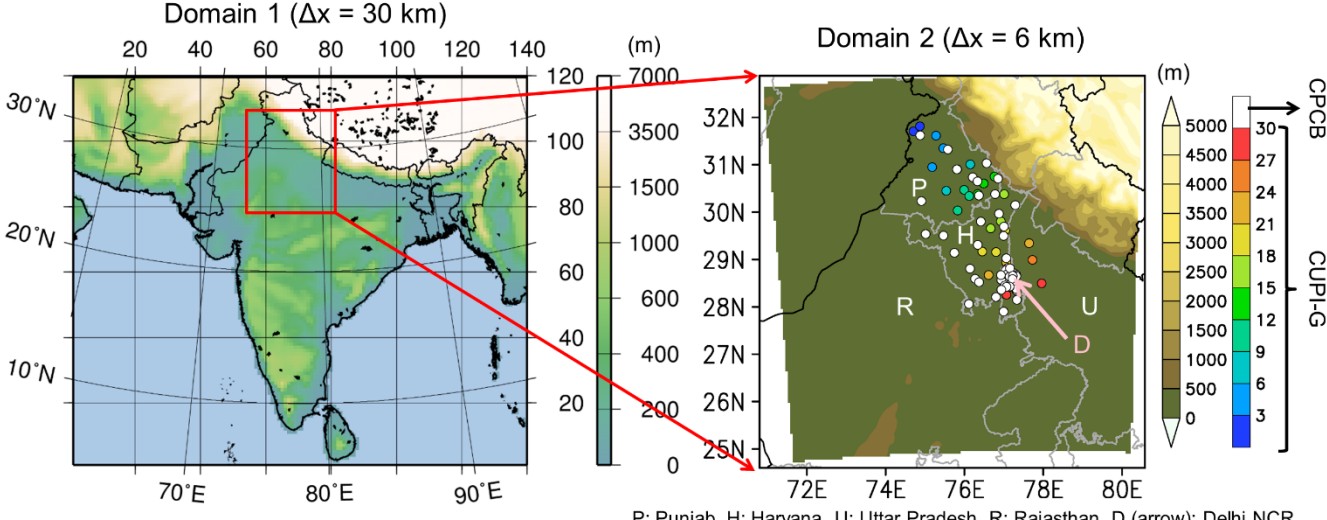

**Figure 1:** Mother model domain (left, $\Delta x$ = 30 km) and nested model domain (right, $\Delta x$ = 6 km) with terrestrial elevations. Circles in the right panel indicate observation sites used in the study, which were provided by the Compact and Useful PM$_{2.5}$ Instrument with Gas Sensors (CUPI-G) observation network (colored circles) and the Central Pollution Control Board (CPCB) in India (white circles). The names of states in India, such as Punjab, Haryana, Uttar Pradesh, and Rajasthan, and the Delhi National Capital Region (NCR) are depicted in the right panel.

Among physics modules in WRF, we employed Morrison's double-moment cloud microphysics scheme (Morrison et al., 2009), the RRTMG-K scheme for shortwave and longwave radiation (Baek, 2017), the Mellor–Yamada–Janjic scheme for planetary boundary layer turbulence (Janjic, 1994), and the Unified Noah Land–Surface Model (Chen and Dudhia, 2001) for both domains in WRF. The Grell–Freitas ensemble scheme (Grell and Freitas, 2014) was utilized for sub-grid scale cumulus parameterization only over D01. For the aerosol and chemistry modules in the CTM of NHM-Chem, we employed the same approach as Kajino et al. (2019, 2021), adopting the 5-category nonequilibrium method for aerosol representation. Within the 5-category method, aerosols are categorized into five categories or modes, including the Aitken mode (ATK), soot-free accumulation mode (ACM), internal mixture of soot aggregates (AGR), mineral and anthropogenic dust (DU), and sea salt particles (SS), and changes in the size distribution and chemical components in each category due to emissions, secondary production, new particle formation, advection, turbulent diffusion, and dry and wet deposition processes are solved dynamically using a triple-moment modal dynamics approach (Kajino, 2011).

As demonstrated in Fig. 2, we considered anthropogenic and natural emissions for the CTM simulation. We employed the Reginal Emission inventory in Asia (REAS) version 3.2.1 (monthly, $\Delta x$ = 0.25°, base year = 1950–2015) (Kurokawa and Ohara, 2020; 2021) as the anthropogenic emissions for D01 and D02. We utilized 2015 values for the simulation because 2015 was the latest dataset available. The near-real-time open biomass burning emissions provided by



the Global Fire Assimilation System (GFAS; daily, $\Delta x = 0.1°$, 2003 to present) version 1.2 (Kaiser et al., 2012) were utilized for the simulation of D02. Note that fire emissions were not considered in the D01 simulation because we assumed that fire emissions from outside D02 had little influence on surface $PM_{2.5}$ concentrations over the targeted region during the simulation period. For natural sources, we employed the scheme of Han et al. (2004) for mineral dust deflation, the scheme

of Clarke et al. (2006) for sea salt emissions, and the Model of Emissions of Gases and Aerosols from Nature (MEGAN2; Guenther et al., 2006) for biogenic emissions. No volcanic $SO_2$ emissions were considered since there were no active volcanoes in the region. The hourly and vertical profiles of Li et al. (2017) were applied for the anthropogenic emissions of REASv3.2.1. Natural emissions, such as mineral dust, sea salt, and biogenic compounds, which vary in time, were derived from simulated hourly surface meteorological variables. Open biomass burning emissions (GFAS) were assumed to be

constant over time, and their vertical distribution was assumed to be uniform up to 1 km above ground level, based on the data described in Tang et al. (2022). The emissions of $NO_x$, $SO_2$, $NH_3$, nonmethane volatile organic compounds (NMVOC), black carbon (BC), primary organic carbon (OC), primary $PM_{2.5}$, and primary $PM_{10}$ were utilized in the simulation, as described by Kajino et al. (2021). However, only carbon monoxide (CO) data are depicted in Fig. 2 to illustrate the spatial variations of anthropogenic and open biomass burning emissions in the study area.

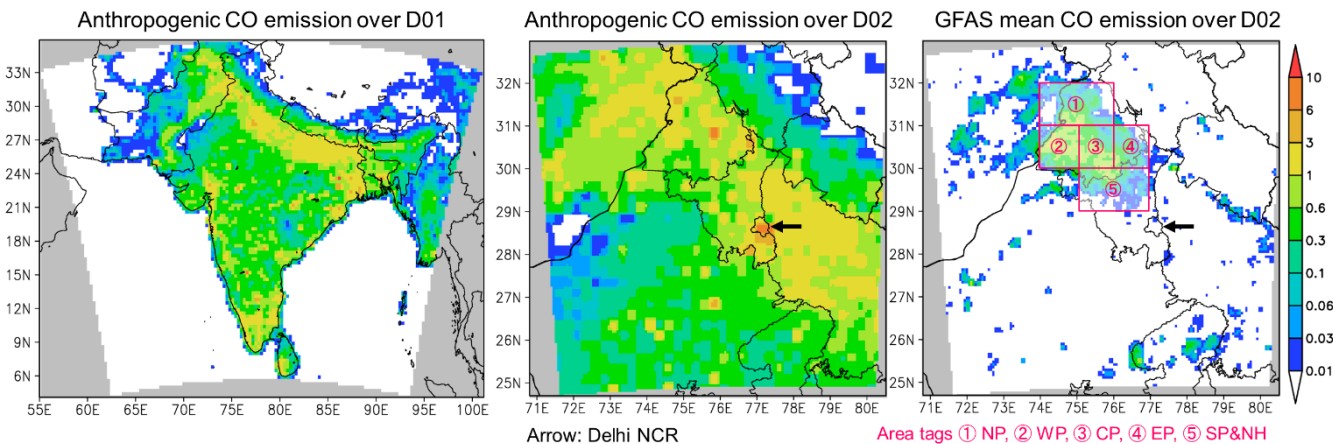

**Figure 2:** (left) Anthropogenic CO emissions (μg m$^{-2}$ s$^{-1}$) of November 2015, provided by REASv3.2.1 over D01. (center) Same as (left) but over D02. (right) Same as (center) but the simulation period-mean open biomass burning CO is provided by the Global Fire Assimilation System (GFAS). The black arrows in the center and right panels indicate Delhi NCR. The pink areas in the right panel indicate the five regions for area tags, North Punjab (NP), West Punjab (WP), Central Punjab

(CP), East Punjab (EP), and South Punjab and North Haryana (SP&NH).



## 2.2 Observational data

### 2.2.1 Monitoring data provided by CPCB

CPCB provides near-real-time surface monitoring data of air pollutants and meteorological variables from CAAQM stations. There are currently 542 stations across India (https://airquality.cpcb.gov.in/ccr/#/caaqm-dashboard-all/caaqm-landing, last accessed: May 7, 2024) providing 15 min–averaged data of air pollutants such as CO, $SO_2$, NO, $NO_2$, $NO_x$, $O_3$, $PM_{2.5}$, $PM_{10}$, benzene, toluene, xylene, ethyl benzene, m-xylene, p-xylene, methane ($CH_4$), $NH_3$, HCHO, and Hg , as well as meteorological variables such as temperature, wind speed, wind direction, relative humidity, pressure, solar radiation, and rainfall. The technical documentation can be found at https://erc.mp.gov.in/Documents/doc/Guidelines/CAAQMS_Specs_new.pdf (last accessed: May 7, 2024), and raw data are available at https://app.cpcbccr.com/ccr/#/caaqm-dashboard/caaqm-landing/ (last accessed: May 7, 2024). Former 1 h–averaged data based on UTC were derived from 15 min–averaged data to compare with the simulation data. We obtained the observational data available in 2022 from 40, 30, and 8 stations in Delhi National Capital Region (NCR), Haryana state, and Punjab state, respectively, as listed in Table S1.

### 2.2.2 High-density in situ surface observation network using CUPI-G stations

The CUPI-G field campaign was described by Singh et al. (2023) and details of the low-cost $PM_{2.5}$ sensors were presented by Nakayama et al. (2018); however, some key features are included here. A CUPI-G station consists of a low-cost $PM_{2.5}$ sensor developed by Panasonic Co., Ltd. (Osaka, Japan) (Nakayama et al., 2018) and other low-cost sensors for gases provided by AMETEK Inc. (Berwyn, PA, USA), such as the CO-B4 Carbon Monoxide Sensor, NO-B4 Nitric Oxide Sensor, NO2-A43F Nitrogen Dioxide Sensor, and the OX-A431 Oxidizing Gas Sensor (https://www.alphasens.com; last accessed: May 7, 2024). The 15 min–averaged data were generated based on the 2 min raw data after a quality check, and then the former 1 h–averaged data based on UTC were derived from 15 min-averaged data to compare with the simulation data. There were 29 stations available in 2022. As listed in Table S1, Singh et al. (2023) categorized the CUPI-G stations into "source," "intermediate," and "Delhi NCR" regions, which were basically based on state boundaries, namely, Punjab, Haryana, and Delhi NCR, and some CUPI-G stations from other states. The site names, locations, and location types can be found in the supplementary material of Singh et al. (2023) and the same category definitions were adopted herein. All 14 stations in Punjab were categorized as the source region. Nine stations were categorized in the intermediate region, including eight stations from Haryana and one station from Uttar Pradesh (Muzaffarnagar). Delhi NCR included six stations, including one station in New Delhi (Jawaharlal Nehru University), two stations in Uttar Pradesh (Meerut and Aurangabad), and three stations in Haryana (two in Gurugram and one in Faridabad). Notably, the categorizations differed for CPCB and CUPI-G stations because CPCB stations were categorized on a state basis.





*2.2.3* Remote sensing data, MODIS, TROPOMI, and AERONET

The Level 3 daily global 1° × 1° aerosol optical depth (AOD) at a wavelength of 550 nm and cloud fraction (CF) data of MODIS aboard the National Aeronautics and Space Administration (NASA) Terra satellite (observation time: 10:30 local time, descending mode) were employed in this study. The AOD_550_Dark_Target_Deep_Blue_Combined_Mean and the Cloud_Fraction_Mean variables of Collection 6.1 were used for the AOD and CF data, respectively. The data description is available at https://atmosphere-imager.gsfc.nasa.gov/sites/default/files/ModAtmo/L3_ATBD_C6_C61_2019_02_20.pdf (last accessed: May 7, 2024).

The 1-orbit Level 2 (5.5 km × 3.5 km) TROPOspheric Monitoring Instrument (TROPOMI) ultraviolet aerosol index (UVAI) using the 340/380 nm wavelength pair data aboard the European Space Agency (ESA) Sentinel-5 Precursor (S5P) satellite was utilized in this study. The data description is available at https://sentinel.esa.int/documents/247904/2474726/Sentinel-5P-Level-2-Product-User-Manual-Aerosol-Index-product (last accessed: May 7, 2024). Even though MODIS AOD is not retrieved in the presence of clouds, UVAI data can be calculated.

The ground-based measurement data of AOD from the Aerosol Robotic Network (AERONET) were employed in model validation as an independent dataset from the data utilized for emission optimization, i.e., CPCB and CUPI-G. AERONET consisted of 434 stations worldwide (two in the D02) in 2022, imposing standardization of the instrument (Cimel's sun photometer, CE318 series), calibration, and data processing. We employed Level 2.0 (quality-assured) Version 3 AOD at a wavelength of 500 nm for validation of the simulated AOD at the same wavelength by assuming the Maxwell Garnett approximation for nonlight-absorbing and light-absorbing internal mixtures, such as BC and mineral dust. The data and their description are found at https://aeronet.gsfc.nasa.gov/ (last accessed: May 7, 2024).

**2.3 Emission optimization using tagged simulation**

To optimize CRB emission fluxes, we proposed a simple cost function basically following the concept of the equation commonly used for Bayesian synthesis inversion (e.g., Eq. 1 in Baker et al., 2006), which is expressed as follows:

$$f = \sum_{n=1}^{N} \frac{\left(O_n - \left(S_{0,n} + \sum_m x_m (S_{m,n} - S_{0,n})\right)\right)^2}{\sigma_O^2} + + \sum_{m=1}^{M} \frac{(x_m - x_0)^2}{(u_m - l_m)^2},$$

(1)

where $O$ and $S$ are the observed and simulated data, respectively; $N$ and $M$ are the numbers of observational data and sensitivity tests (or tags), respectively; $S_{0,n}$ is a simulation result without CRB emissions (i.e., anthropogenic and natural emissions only); $S_{m,n}$ is a tagged simulation including CRB emissions of the right panel of Fig. 2 (period-mean GFAS





emission) for certain times and areas; $\sigma_O$ is the standard deviation of the observational data; $x_m$ is a variable to optimize the cost function with the upper ($u_m$) and lower ($l_m$) limits; and $x_0$ is the initial condition of $x_m$. Similar to Eq. 1 in Baker et al. (2006), the first and second terms on the right-hand side of the equation represent the deviations between simulations and observations and between optimized and a priori simulations, respectively. Optimization reduced both sets of deviations simultaneously. We employed the limited memory Bryoyden–Fletcher–Goldfarb–Shanno (L-BFGS)-B algorithm (L-BFGS-B; Byrd et al., 1995; Zhu et al., 1997) to minimize the cost function. L-BFGS-B is an extension of the quasi-Newton algorithm L-BFGS (Nocedal, 1980; Liu and Nocedal, 1989), which is utilized to minimize a nonlinear function $f(x)$ subject to $l \leq x \leq u$ (Zhu et al., 1997). This optimizer was commonly used in previous studies of atmospheric chemistry inverse modeling (e.g., Zheng et al., 2018).

The list of sensitivity simulations performed herein is summarized in Table 1. We conducted whole-period simulations without CRB emissions ("No_CRB"), with GFAS emissions ("GFAS"), and using several emissions optimized by Eq. 1 ("Optimized"). Among the "Optimized" emission simulations, as described in Section 3.2, the best estimate was referred to as Optimized Fire Emission v1.0 (OFEv1.0). The whole-period simulations began on October 15, 2022 (00:00 UTC), using the same initial conditions simulated by the "No_CRB" test during October 10–15, 2022 (00:00 UTC), that is, the 5-day spin-up period. The tagged simulations and subsequent emission optimization were separately applied for the two plume periods defined by Singh et al. (2023), including the Plume 1 period in November 2–5, 2022 (00:00 IST; India Standard Time) and Plume 2 period in November 8–12 2022 (00:00 IST). The simulation periods were November 1–6, 2022 (00:00 UTC) and November 6–14, 2022 (00:00 UTC) for Plumes 1 and 2, respectively. As shown in Fig. 2, we set five areal tags over the CRB source region, including North Punjab (NP), West Punjab (WP), Central Punjab (CP), East Punjab (EP), and South Punjab and North Haryana (SP&NH). Based on GFAS, the emission contribution from the five regions to the emissions over D02 was approximately 80% from October 15 to November 15, 2022 (00:00 UTC). In addition to the five areal tags, 7 and 11 temporal tags were set for emission optimization for the Plume 1 and Plume 2 periods, respectively, as shown in Table 1. On each day, the temporal tags were divided into AM (00:00–12:00 UTC or 5:30–17:30 IST) and PM (12:00–24:00 UTC or 17:30–5:30 IST) to elucidate the relative abundance of daytime and nighttime fire ignition.



Table 1: Sensitivity simulations

|  | Simulation period | Emission test names | Temporal tags |
|---|---|---|---|
| Whole period | October 15–November 15, 2022 (00:00 UTC) | No_CRB GFAS Optimized OFEv1.0[d] | n.a. |
| Plume 1 period: November 2–5, 2022 (00:00 IST)[a] | November 1–6, 2022 (00:00 UTC) | Tagged | 7 tags (November 1 PM[b]; November 2 AM[c], PM; November 3 AM, PM; November 4 AM, PM) |
| Plume 2 period: November 8–12, 2022 (00:00 IST)[a] | November 6–14, 2022 (00:00 UTC) | Tagged | 11 tags (November 7 AM, PM; November 8 AM, PM; November 9 AM, PM; November 10 AM, PM; November 11 AM, PM; 12 AM) |

[a]The plume periods defined by Singh et al. (2023)

[b]00:00–12:00 UTC (5:30–17:30 IST)

[c]12:00–24:00 UTC (17:30–5:30 IST)

[d]The best estimation in this study: Optimized Fire Emission v1.0 (OFEv1.0).

In total, there were 36 and 56 tags (areal tags multiplied by temporal tags plus one; GFAS tag) for the Plume 1 and Plume 2 periods, respectively, and solving Eq. 1 in one step was found to be unsuccessful. In addition, there was no prior information regarding the upper and lower limits of $x_m$. Therefore, we applied multi-step optimization with smaller limit

values as follows: Step 1 involved solving Eq. 1 using temporal tags only with $(u_m, l_m) = (2.0, 0.5)$ by summing all areal tagged simulations; Step 2 involved solving Eq. 1 using areal tags only with $(u_m, l_m) = (2.0, 0.5)$ by summing all temporal tagged simulations multiplied by the optimized $x_m$ value of the result of Step 1; and Step 3 was the same as Step 1, but it involved summing all areal tagged simulations multiplied by the optimized $x_m$ value of the result of Step 2. This multi-step optimization was repeated until $0.009 < x_m < 1.001$ was obtained for all $m$ values. We set $x_0 = 1$ for all cases.

Observation site selection is essential for better estimations of CRB emissions. Herein, we only selected observation sites where data were continuously available and reliable for the whole simulation period (i.e., the data did not include sudden gaps and exhibited zero drift). Moreover, we only selected sites where the simulated values agreed well with the observed values during the period not affected by CRB, namely, October 15–28, 2022. The NHM-Chem simulation can only predict spatiotemporal mean concentrations from steady-state emission sources, whereas observational data are affected by

the local environment and sporadic emissions. If the spatial and temporal representativeness of an observation site was small or the site was largely affected by local-scale disturbances, deviations between the simulation and observations would be large. Using all observational data, optimization would reduce these deviations due to differences in spatiotemporal



representativeness between the simulation and observations. All CPCB and CUPI-G sites are listed in Table S1, as well as those selected in this study.

The optimized $x_m$ value should vary depending on the chemical compound. Ideally, the same compound should be utilized for optimization and emission changes. For example, the optimized $x_m$ value obtained from observed and tagged

simulations for CO should be applied to optimize CO emissions. However, quality assurance and control (QA/QC) had only been performed for CUPI-G PM$_{2.5}$ data at the time of the study (Singh et al., 2023) and not all variables were available in CPCB data. Therefore, we applied the same $x_m$ values obtained from the observed and tagged simulations of PM$_{2.5}$ for the optimized fire emissions of all primary precursor species, including NO$_x$, SO$_2$, NH$_3$, NMVOC, BC, OC, PM$_{2.5}$, and PM$_{10}$. Thus, we assumed that the relative magnitude of the emission factors for all primary emission species was consistently

estimated in GFAS emissions, and chemical reactions, transport, and deposition processes were consistently solved by NHM-Chem; therefore, the deviations between "GFAS" simulations and observations only originated from discrepancies in spatiotemporal variations of satellite-derived fire detection over the region and difference in emission factors which is common for all chemical compounds. Although this assumption was unlikely, it was the best available option.

## 3. Results and discussion

### 3.1 Time series comparisons

Fig. 3 presents the time series of station-mean observed and simulated emissions by "GFAS" for CPCB and CUPI-G stations over Punjab (source), Haryana (intermediate), and Delhi NCR. The observational data differed for CPCB and CUPI-G stations on October 24, 2022, which was the date of Diwali, the festival of lights. Because CPCB stations are situated in urban locations, remarkable enhancements in PM$_{2.5}$ were observed at Punjab and Haryana CPCB stations due to

festive use of firecrackers and candles, whereas no enhancements were observed at CUPI-G source and intermediate stations situated in farmland areas. Because the model did not consider sporadic emissions due to events such as Diwali, enhancements were not simulated on October 24, 2022 in these regions. Thus, optimization using CPCB data that included the Diwali day would cause misleading CRB emissions. Another remarkable feature was the large difference in observed PM$_{2.5}$ between the CPCB Punjab and CUPI-G source regions. The observed and simulated PM$_{2.5}$ for CPCB Punjab did not

differ greatly, except on November 8 and 9, whereas large underestimations were found in the simulation for the CUPI-G source region from October 29 to November 10, 2022, which was the intensive CRB period of that year. A similar feature was observed in CPCB Haryana and CUPI-G intermediate regions, where observation values at CUPI-G stations over the CRB period were larger than those observed in CPCB Haryana and those simulated by "GFAS". These features demonstrated the successful allocation of CUPI-G stations over source and intermediate regions, thereby avoiding large

anthropogenic emission sources and capturing the influence of CRB emissions.



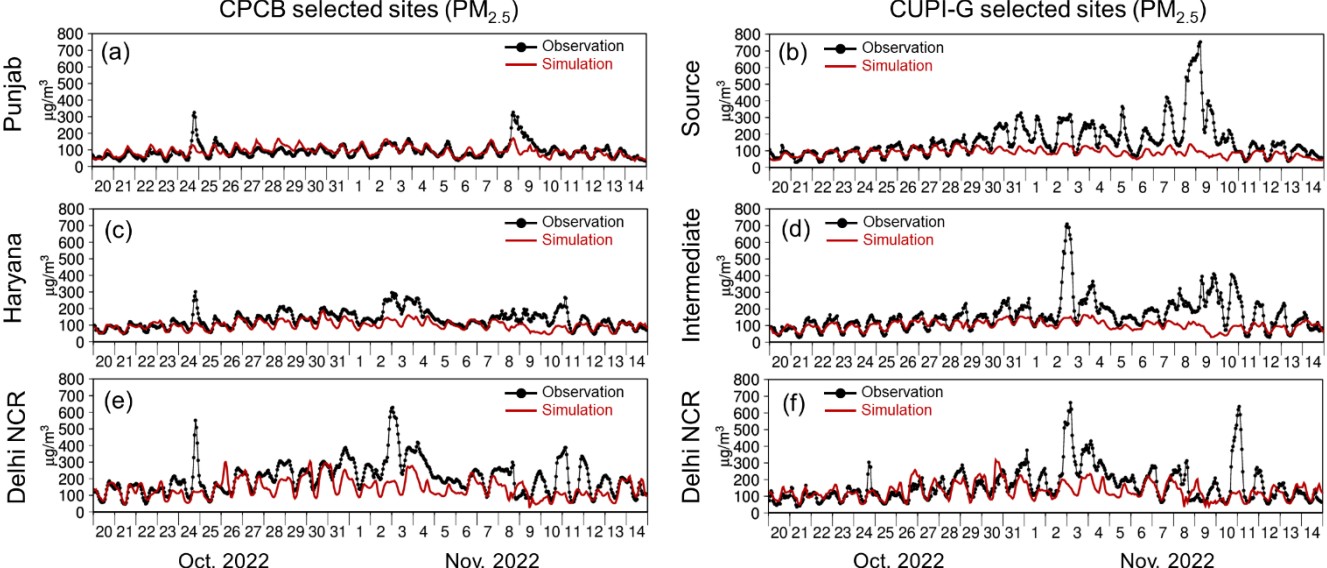

**Figure 3:** Temporal variations (in UTC) of selected station-mean (black) observed and (red) "GFAS" simulated PM$_{2.5}$ concentrations (μg m$^{-3}$) for (a) CPCB Punjab, (b) CUPI-G Source, (c) CPCB Haryana, (d) CUPI-G Intermediate, (e) CPCB Delhi NCR, and (f) CUPI-G Delhi NCR from October 20 to November 15, 2022 (00:00 UTC).

The time series differences between selected and all stations for PM$_{2.5}$ and CO are presented in Figs. S2 and S3, respectively. The importance of station selection was evident, especially for CO in Delhi NCR (Fig. S3). The "GFAS" simulation seemed to underestimate emissions due to the underestimation of CRB emissions only for CPCB selected stations over Delhi NCR (Fig. S3e), whereas the simulation tended to underestimate emissions almost every night for all CPCB stations over Delhi NCR (Fig. S3f). This was likely because CPCB stations were located within the urban canopy, whereas the simulation predicted air concentrations above it. The inclusion of all CPCB station data in emission optimization would prevent the optimization of only the effect of CRB emissions. The same trends were observed in Punjab and Haryana. However, the PM$_{2.5}$ differences between selected and all stations (Fig. S2) were not overly remarkable compared to those of CO, which was likely because CO is generated by primary sources, whereas PM$_{2.5}$ consists of both primary and secondary sources. Concentrations within and above the canopy for primary species such as CO may differ substantially because emission contributions dominate air concentrations within the canopy. However, for secondary species, the contribution of transport from other regions became relatively larger, thereby reducing concentration differences within and above the canopy. The observed PM$_{2.5}$ was slightly greater in all stations than in selected stations, especially for the CUPI-G intermediate region (Fig. S2i, j); however, the differences were minor compared to those of CO.

Fig. 4 presents the observed and simulated time series of selected station-mean meteorological variables (the selection of stations was based on PM$_{2.5}$ data, not CO data). Since no height information was available for the wind measurements of CPCB, height adjustments were not applied in the figure. The simulated 10 m wind data were compared

ehttps://doi.org/10.5194/egusphere-2024-1811




with the observed data, which seemed to be <10 m above ground level or within the urban canopy, as previously discussed, because the simulation yielded substantially larger values than the observations. Nevertheless, good agreement was achieved between the simulation and observations in terms of temporal variations for wind speed, except on November 8 and 9, 2022, which was likely due to clouds and precipitation, especially over Delhi. As wind direction is affected by the local

5    environment and mesoscale meteorology, the derivation of station-mean wind direction was not overly meaningful. Nevertheless, the simulated wind direction matched well with observations over Haryana, where northwesterly winds prevailed over the Plume 1 (November 2–4, 2022) and Plume 2 (November 8–12, 2022) periods. In Delhi NCR, the simulated winds were northwesterly during the two plume periods, whereas those in observations were southwesterly. This was likely due to many of the selected stations being located within the urban canopy, causing the simulated wind direction

10    to deviate from observed values. This difference in wind direction in Delhi NCR may have caused errors in emission optimization, but the effect could not be quantitatively derived. The observed wind directions in Punjab during the two plume periods were almost northwesterly. Similarly, the simulated values were almost northwesterly, although they sometimes deviated from observations, especially in the early stage of Plume 2 (November 8–9, 2022).






**Figure 4:** Same as Fig. 3 but for selected CPCB station-mean (top left; a, c, e) wind speed (m s⁻¹), (top right; b, d, f) wind direction (degree), (bottom left; g, i, k) temperature (°C), and (bottom right; h, j, l) relative humidity (%, left axis) over Punjab, Haryana, and Delhi NCR, respectively. The simulated (blue lines) and observed (sky blue bars) surface precipitation amounts (mm h⁻¹) are also depicted in the bottom right panels on the right axis (h, j, l).





Figs. 5 and 6 presents the time series of spatial distributions of GFAS CO emissions used in the simulation, MODIS AOD and CF, and TROPOMI UVAI during November 1–5 and 8–12, 2022, which included the Plume 1 and Plume 2 periods, respectively. The same data for the entire CRB period (October 27 to November 15, 2022) are presented Figs. S4–S7, as well as the simulated surface concentrations of anthropogenic PM$_{2.5}$ and OFEv1 CRB PM$_{2.5}$ and wind vectors. Notably, the data in the figures were obtained around noon, but the timing differed slightly: approximately 10:30 IST for MODIS, 12:00–15:00 IST for TROPOMI, and 11:30 IST for the simulation. Additionally, MODIS and TROPOMI represent column values, whereas the simulation data are in the surface air (bottom layer of the model grids).

**Figure 5:** Spatial distribution of (top to bottom) daily mean GFAS CO emissions, MODIS AOD, MODIS Cloud Fraction, and TROPOMI UV Aerosol Index during November 1–5, 2022, which includes the Plume 1 period (November 2–4, 2022). The Terra descending mode data (10:30 local time) is shown for MODIS. The time periods of TROPOMI orbit data used for the figure are: 7:35–9:16, 7:16–8:57, 6:56–8:38, 6:37–8:19, and 6:18–8:00 UTC for November 1–5, 2022, respectively.





**Figure 6:** Same as Fig. 5 but during November 8–12, 2022, which includes the Plume 2 period (November 8–12, 2022). The time periods of TROPOMI orbit data used for the figure are: 7:02–8:44, 6:43–8:25, 6:24–8:06, 7:47–9:28, and 7:28–9:09 UTC for November 8–12, 2022, respectively.

During the Plume 1 period (Figs. 5 and S5), remarkable signs of CRB emissions were observed over Punjab due to substantial underestimations of simulated $PM_{2.5}$ in the CUPI-G source region data, as shown in Fig. 3b. There were higher GFAS emissions on November 2 and 4, 2022, corresponding to large values of MODIS AOD (>2). MODIS retrieval was unsuccessful on November 3 and 5, 2022, likely due to the presence of clouds, thus GFAS emissions were small, whereas TROPOMI UVAI indicated a substantially large aerosol burden over the PHD region (>3). Southeasterly winds prevailed on November 1 and 5, 2022 (Fig. S5), preventing transportation of surface $PM_{2.5}$ from CRB to Haryana and Delhi (Fig. 3). However, from November 2 to 4, 2022, northwesterly winds carried air pollutants from Punjab to Delhi, which caused high surface $PM_{2.5}$ concentrations of >500 μg m$^{-3}$. In fact, the Plume 1 period could be divided into two events, including Plume 1A on November 2–3, 2022 and Plume 1B on November 3–4, 2022. The Plume 1A event (station mean-values =





approximately 600–700 µg m$^{-3}$) was larger than the Plume 1B event (approximately 300–400 µg m$^{-3}$) (Fig. 3d–f), likely because the former plume directly transported pollutants from Punjab to Delhi, while the latter plume was a blowback of pollutants previously carried downwind (southeast direction) of Delhi by the former plume (Figs. 4d, f, and S5).

Similarly, the Plume 2 period could be divided into two events, including Plume 2A on November 8–9, 2022 and Plume 2B on November 10–12, 2022. During the Plume 2 period (Figs. 6, S6, and S7), remarkable signs of CRB emissions on November 8–9, 2022 were observed in Punjab, as the CUPI-G (farmland) mean PM$_{2.5}$ concentrations were >700 µg m$^{-3}$ (Fig. 3b) and even the CPCB (urban) mean values became much higher (300 µg m$^{-3}$) (Fig. 3a) than the GFAS simulation (100 µg m$^{-3}$) (Fig. 3a, b). However, MODIS AOD data were not available during this period, and even TROPOMI UVAI did not detect a high concentration event in Punjab on November 8, 2022, likely due to thick clouds associated with rainfall over the PHD region (Fig. 4h, j, l). CRB cannot be conducted in the rain; however, CRB might have occurred because rain was not observed in Punjab during this period (Fig. 4h). The surface PM$_{2.5}$ concentrations in Punjab were high on November 8–9, 2022, whereas those in Delhi NCR were high on November 10–12, 2022 (Fig. 3b, f). However, unlike during the Plume 1 period, there were no remarkable enhancements of MODIS AOD and TROPOMI UVAI over Punjab on November 8–9, 2022 and Delhi NCR on November 10–12, 2022 during the Plume 2 period (Fig. 6). In the latter period, the transport of air pollutants from Punjab to Delhi was observed in the simulation (Fig. S7). By contrast, CRB plumes were not transported to the Haryana region in the former period (Fig. S6), whereas an enhancement was observed in surface PM$_{2.5}$ at CUPI-G intermediate stations (Fig. 3d). As shown in Fig. 4a–f, discrepancies between simulated and observed wind fields were enhanced in the former period (November 8–9, 2022), especially for Delhi NCR (Fig. 4e, f). The meteorological simulation might have failed to reproduce the air flows associated with rainfall from thick convective clouds, hindering successful optimization of CRB emissions during this period.

## 3.2 Optimization of CRB emissions

Various combinations of data selection, application, and evaluation could be proposed for CRB emission optimization, including optimization using CUPI-G data and validation using CPCB data. Another approach could be emission optimization utilizing data from Punjab and validation using data from Delhi. We needed emission data of all chemical components for the simulation of PM$_{2.5}$; however, not all data were available from observations, as described in Section 2.3. We attempted various combinations of data selection and found that employing more data yielded better results; therefore, we decided to use PM$_{2.5}$ data from the CPCB and CUPI-G stations over all regions as an optimization starting point.

Table 2 summarizes the data selection combinations employed in optimization and the obtained $x_m$ values from Eq. 1. "CPCB + CUPI-G" indicates optimization using PM$_{2.5}$ data from all CPCB and CUPI-G stations. Optimization was performed for each plume period, so that different $x_m$ values were obtained for the two plume periods. Because the simulated



wind fields deviated from observations in the Plume 2 period (November 8–9, 2022), $x_m$ for the GFAS tag during the Plume 1 period (6.87 or 6.99) was applied for the whole simulation period (October 15–November 15, 2022 (00:00 UTC)). For the areal tags, only time-averaged data are presented in Table 2; however, $x_m$ values were also obtained for different temporal tags, which were applied for each period. The time series of observed and simulated all and selected CPCB and CUPI-G station-average PM$_{2.5}$ concentrations are presented in Fig. 7. The simulated source contributions are also indicated in the lower half of the same figure. The anthropogenic contribution was calculated using the brute-force method, derived from the difference between control and sensitivity runs with the source's emission removed (zeroed out) or the deviation from the difference between the control run and the 20% emission reduction run multiplied by 5 to reduce the effect of nonlinearity in chemical reactions (the zeroed-out simulation may affect oxidant concentrations and thus chemical production rates of secondary PM$_{2.5}$, which deteriorates the calculation of source contribution estimations). The brute-force method is easy to implement, and its validity has been confirmed via comparisons with other sophisticated methods, such as the decoupled direct method (Hakami, 2004; Choi et al., 2014; Kelly et al., 2015). The brute-force method with a 20% reduction run was employed to obtain the anthropogenic PM$_{2.5}$, whereas the zeroed-out brute-force method was applied to obtain PM$_{2.5}$ from CRB emissions. The miscellaneous contribution was the remainder after subtracting all contributions, namely, natural emissions, upper and lateral boundary effects, and numerical errors.

Table 2: Combination of observational data selection utilized for emission optimization and obtained $x_m$ values.

| | Data selection | GFAS tag | Areal tags | | | | |
|---|---|---|---|---|---|---|---|
| | | | 1. NP | 2. WP | 3. CP | 4. EP | 5. SP&NH |
| Plume 1 period | CPCB + CUPI-G[a] | 6.87 | 0.00 | 0.00 | 0.00 | 0.00 | 31.6 |
| | CUPI-G Punjab[b] | 6.99 | 0.00 | 0.00 | 0.00 | 38.9 | 0.00 |
| | CUPI-G P89[c] | - | - | - | - | - | - |
| Plume 2 period | CPCB + CUPI-G[a] | 3.03 | 0.00 | 0.00 | 0.00 | 47.7 | 9.84 |
| | CUPI-G Punjab[b] | 4.13 | 0.00 | 0.00 | 9.80 | 69.1 | 0.00 |
| | CUPI-G P89[c] | 2.09 | 0.00 | 0.00 | 7.63 | 116 | 0.00 |

[a]Optimization using all CPCB and CUPI-G stations.
[b]Optimization using selected CUPI-G Punjab stations.
[c]Optimization using only CUPI-G stations No. 8 (Thikriwala) and No. 9 (Beauscape Farm) in Punjab state.





**Figure 7:** (a–f) Time series of all and selected CPCB and CUPI-G station-average (black) observed and simulated (red) total and (blue) No_CRB PM$_{2.5}$ using (left) GFAS emission and (right) optimized fire emission (CPCB + CUPI-G) over (top to bottom) source (CPCB Punjab + CUPI-G Source), intermediate (CPCB Haryana + CUPI-G Intermediate), and Delhi NCR (CPCB Delhi NCR + CUPI-G Delhi NCR) regions from November 1 to 5, 2022 (00:00 UTC). (g–l) Same as (a–f) but for simulated contributions of different emission sources, (gray) anthropogenic emissions, CRB emissions from (red) North Punjab, (pink) West Punjab, (yellow) Central Punjab, (green) East Punjab, (sky blue) South Punjab and North Haryana, (blue) fire from other areas, and (white) miscellaneous, which contains natural emissions such as biogenic, mineral dust, and sea salt, influences from lateral and upper boundary conditions, and numerical errors.



The "GFAS" simulation substantially underestimated surface $PM_{2.5}$ concentrations over the entire PHD region (Fig. 7a, c, e); however the simulation using CRB emissions optimized by "CPCB + CUPI-G" data (Fig. 7b, d, f) was substantially improved, especially for the Plume 1 (November 2–4, 2022) and Plume 2B (November 10–12, 2022) periods, demonstrated by increased emissions in the EP and SP&NH regions. GFAS emissions are generally smaller than CRB fire emissions in India (Cusworth et al., 2018; Wiedinmyer et al., 2023). Moreover, the GFAS emissions may have been underestimated in our case because the optimized $x_m$ values were much larger than 1.

Even though CRB emissions optimized by "CPCB + CUPI-G" data substantially improved the simulation of $PM_{2.5}$, the simulation continued to greatly underestimate the observed peaks in Plume 2A (8–9 November) over the entire PHD region. Therefore, additional optimizations were conducted using CUPI-G Punjab data only. The "CPCB + CUPI-G" optimization was insensitive to emissions from the NP, WP, and CP regions ($x_m < 10^{-5}$), whereas the additional optimization "CUPI-G Punjab" enhanced emissions from CP. Because the additional optimization continued to underestimate surface concentrations, especially on November 8 and 9, 2022, an additional optimization was conducted using only data from CUPI-G station numbers 8 and 9 ("CUPI-G P89"), where substantial enhancement of $PM_{2.5}$ was observed on these dates. Subsequently, we constructed two additional optimized CRB emission inventories in the same manner as the "CPCB + CUPI-G" case, which were merged with "CPCB + CUPI-G" emissions by taking the larger $x_m$ values among CPCB + CUPI-G and the additional cases, regarded as "CPCB + CUPI-G plus CUPI-G Punjab" and "CPCB + CUPI-G plus CUPI-G P89." Note that optimization of "CUPI-G P89" was only conducted for the Plume 2 period and the same $x_m$ values with "CUPI-G Punjab" were used for the Plume 1 period.

The two additional simulation results obtained using "CPCB + CUPI-G plus CUPI-G Punjab" and "CPCB + CUPI-G plus CUPI-G P89" emissions are shown in the left and right columns of Fig. 8, respectively. Simulated $PM_{2.5}$ in the source region was improved due to enhanced emissions over the CP region on November 8–9, 2022 (Fig. 8a, b). However, no increases in the simulation were observed over the intermediate region and Delhi NCR in the same period (Fig. 8c–f), likely because the simulated wind fields deviated from those observed during this period, as shown in Fig. 4. Consequently, the reasons for the enhanced $PM_{2.5}$ concentrations observed over the intermediate region and Delhi NCR on November 8–9, 2022 were not identified in the current study. Statistical scores between the simulated and observed surface concentrations are listed in Table 3. Substantial improvements were achieved in the simulations using optimized emissions compared to the original simulation using GFAS, as demonstrated by the simulation to observation median ratio (*Sim./Obs.*) and correlation coefficient (*R*). Only small differences were observed among the three simulations using optimized emissions, but "CPCB + CUPI-G plus CUPI-G Punjab" was slightly better than the other simulations over the source region, which is referred to as OFEv1.0. Comparison against AERONET AOD at the Lahore and Amity University Gurgaon stations, as shown in Fig. S8, supported the same conclusion: the optimized emissions were better than GFAS, with no great differences observed among the optimized simulations.





**Figure 8:** Same as Fig. 7 but for simulations using (left) optimized fire emission (OFE v1.0; average of CPCB + CUPI-G and CUPI-G Punjab sites) and (right) optimized fire emission (CPCB + CUPI-G and CUPI-G station numbers 8 and 9 in Punjab).





Table 3: Statistical scores comparing simulated and oserved PM$_{2.5}$ over the PHD region using several CRB emission cases.

| Fire emissions | Source | | Intermediate | | Delhi NCR | |
|---|---|---|---|---|---|---|
| | Sim./Obs.[a] | R[b] | Sim./Obs. | R | Sim./Obs. | R |
| GFAS | 0.67 | 0.59 | 0.71 | 0.50 | 0.61 | 0.47 |
| CPCB + CUPI-G | 0.98 | 0.60 | 0.91 | 0.54 | 0.85 | 0.64 |
| CPCB + CUPI-G plus CUPI-G Punjab[c] | 1.02 | 0.71 | 0.92 | 0.57 | 0.85 | 0.64 |
| CPCB + CUPI-G plus CUPI-G P89 | 1.03 | 0.69 | 0.92 | 0.58 | 0.85 | 0.64 |

[a]Simulation to observation median ratio

[b]Correlation coefficient ($R$)

[c]Optimized Fire Emission ver1.0 (OFEv1.0)

During the first half of November 2022, as shown in Figs. 7 and 8, even though its magnitude was small, the contribution of CRB emissions was continuously observed in the "GFAS" simulation in the source region (Fig. 7g), whereas the simulated PM$_{2.5}$ was only affected during the two plume events in the intermediate region and Delhi NCR (Fig. 7i, k). After emission optimization using "CPCB + CUPI-G," approximately 50% of surface PM$_{2.5}$ concentrations accounted for

CRB emissions in the entire PHD region during the Plume 1 period (November 2–3, 2022). In the intermediate region and Delhi NCR (Fig. 7j, l), contributions from SP&NH were substantially increased during the Plume 1A event (November 2–3, 2022), whereas contributions from EP were also increased during the Plume 2B event (November 3–4, 2022). In the source region, CRB emissions from CP and EP contributed almost equally to simulated PM$_{2.5}$. Even though the transport of pollutants by northwesterly winds was clearly observed during the Plume 1 period, the largest contributions to PM$_{2.5}$ in the

source (Punjab) and receptor (Haryana and Delhi NCR) regions differed from one another. After additional inclusion of CUPI-G Punjab stations into the emission optimization (Fig. 8), the contributions from CP were enhanced during the Plume 2A event (November 8–9, 2022) only in the source region, and the CRB emission contributions were >80%. Although the observed concentrations in the receptor region became higher during the Plume 2A event, the simulation did not present any increases (Fig. 8i–l), likely due to the large discrepancy with simulated wind fields, as previously mentioned. During the

Plume 2B event (November 10–12, 2022), in the presence of prevailing northwesterly winds in the simulation, PM$_{2.5}$ in the receptor regions was substantially affected by CRB emissions from CP on November 10–11, 2022 (approximately 50% of total PM$_{2.5}$) and WP, EP, and SP&NH on November 11–12, 2022 (Fig. 8i–l). The CRB emission contributions were >80% on November 11–12, 2022 in receptor regions (Fig. 8i–l). Similar to the Plume 1 period, the largest contributions to PM$_{2.5}$ in the source (Punjab) and receptor (Haryana and Delhi NCR) regions differed from one another during the Plume 2B event,

with PM$_{2.5}$ greatly affected by emissions from NP and EP in the source region and emissions from WP, EP, and SP&NH greatly affecting the receptor region.



Fig. 9 illustrates the time series of D02 total 12 h CRB CO emissions for GFAS and OFEv1.0. The OFEv1.0 to GFAS CO emission ratio was approximately 7 (Table 2), and the two datasets are separately depicted on the left and right axes. Although the $x_m$ values had almost no limit because optimization was repeatedly applied until the values converged, the time variation of OFEv1.0 was continuous without any sudden spikes or gaps. This indicated that the number of

observational data available was sufficient for the number of tags. The optimization was successful for the recovery (or boosting) of emission fluxes, especially on November 8 and 10, 2022, when GFAS emissions may have been underestimated due to clouds or thick haze. Large differences were sometimes observed between optimized emissions in daytime (or AM in UTC) and nighttime (or PM. in UTC), such as on November 8, 2022. Because each CRB event lasted only a few hours, the 12 h time resolution may not have been sufficient to reproduce CRB activity. Nevertheless, the study findings indicate the

importance of diurnal variations in CRB emission estimations for air quality simulations. In addition, diurnal variation shape may not be the same on each day. For example, daytime (AM) emissions were larger on November 2, 9, and 10, 2022, while nighttime (PM) emissions were larger on November 3, 4, and 8, 2022. Judging from Fig. 5, GFAS emissions may have been underestimated due to the presence of clouds on November 3, 2022; however, OFEv1.0 did not demonstrate a substantial increase in emissions on this day (Fig. 9). This was likely because the observation stations were insensitive to CRB

emissions that occurred on November 3, 2022 or not many CRB events occurred on that day.

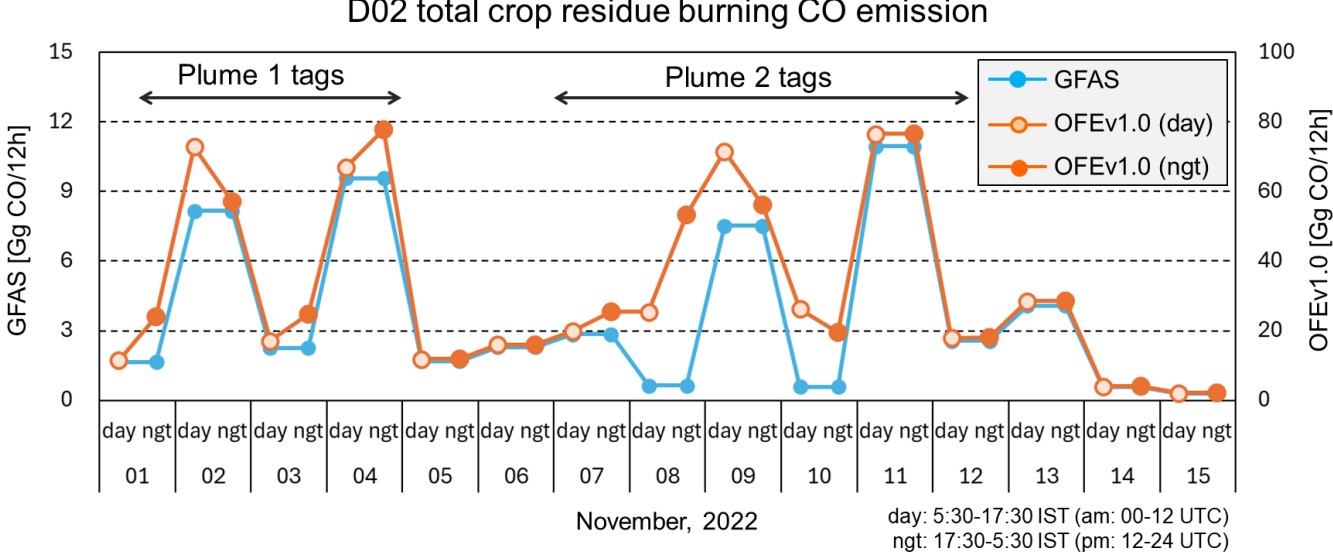

**Figure 9:** Time series of D02 total 12 h CRB CO emissions (Gg CO 12h$^{-1}$) of (sky blue, left axis) GFAS and (orange, right

axis) OFEv1.0 from 1 to 15 November 2022 (in UTC). Time resolution of GFAS is 1 day, whereas OFEv1.0 is a 0.5 day. Daytime (day; 5:30–17:30 IST; or AM, 00:00–12:00 UTC) and nighttime (ngt; 17:30–5:30 IST; or PM, 12:00–24:00 UTC) are separately marked as open and closed circles, respectively.



### 3.3 Intercomparison of OFEv1.0 and other studies

Table 4 compares the emissions of OFEv1.0 and previous studies developed for post monsoon CRB emissions in North India on a total amount basis. The total GFAS emissions from November 1 to 15, 2022 (00:00 UTC) for CO, PM$_{2.5}$, OC, and BC were 111, 10.0, 5.04, and 0.517 Gg, whereas those of OFEv1.0 were 963, 86.1, 43.5, and 4.46 Gg, respectively. The OFEv1.0 emissions were 8.6 times larger than those of GFAS (same ratio for all chemical compounds, as previously explained). The daytime and nighttime emissions of OFEv1.0 did not differ considerably; for example, nighttime CO emissions (495 Gg) were 5.8% larger than daytime CO emissions (468 Gg). The OFEv1.0 to GFAS ratio for the non-tagged period (6.99 in Table 2) can be regarded as the general underestimation of GFAS on clear sky days. A recent study (Wiedinmyer et al., 2023) reported that GFAS emissions may be approximately one order of magnitude smaller than those of Fire Inventory from National Center for Atmospheric Research version 2.5 (FINNv2.5), especially for South and Southeast Asia (Fig. 4 of Wiedinmyer et al., 2023), which was consistent with our general underestimation ratio of 6.99. The difference between the 14-day total OFEv1.0 to GFAS ratio (8.6 times) and the general underestimation ratio (8.6 / 6.99 = 23%) can be regarded as the boosting of emissions that could not be detected due to clouds or thick haze. Liu et al. (2020) estimated that post monsoon CRB emissions were boosted by 142% considering cloud/haze gap fill, which was substantially larger than our estimates.

Emission optimization was performed for November 1–15, 2022 (00:00 UTC); however, applying the same general underestimation ratio (6.99) to boost the emissions from October 15 to November 1, 2022 (00:00 UTC), yielded the monthly total emissions of OEFv1.0 (referred to as OFEv1.0*) (Table 4). The OFEv1.0* monthly total optimized emissions from October 15 to November 15, 2022 (00:00 UTC) were 1460, 130, 65.8, and 6.73 Gg for CO, PM$_{2.5}$, OC, and BC, respectively. Based on GFAS, the monthly values accounted for >85% and approximately 90% of the whole post monsoon emission period (from October 1 to December 1, 2022 (00:00 UTC)) over the D02 and CRB source regions (areal tagged regions), respectively. The monthly total emissions of OFEv1.0* were consistent with those reported in previous studies, such as by Liu et al. (2020), at 65 ± 18 and 5.6 ± 1.6 Gg for OC and BC, respectively, and Beig et al. (2020), at 141.65 Gg for PM$_{2.5}$, but did not align with CO estimated by Liu et al. (2020), at 791 ± 225 Gg. Our emissions may have been overestimated because the same optimized factor obtained using PM$_{2.5}$ data was applied for all species. Certainly, optimization using observed and simulated CO data would be better in CO emission estimations. In fact, differences between observed and simulated CO during the Plume 1 and 2 events (Fig. S3a, c, e) is much smaller than those of PM$_{2.5}$ (Fig. 3a, c, e). Therefore, CO of GFAS may not be underestimated to the extent of PM$_{2.5}$.



Table 4: Total emission amounts (Gg) of post monsoon CRB emissions from North India estimated in this study and previous studies.

| | Period | Duration | CO | PM$_{2.5}$ | OC | BC |
|---|---|---|---|---|---|---|
| OFEv1.0 | November 1–15, 2022[b] | 14 days | 963 | 86.1 | 43.5 | 4.46 |
| OFEv1.0*[a] | October 15–November 15, 2022[b] | 1 month | 1460 | 130 | 65.8 | 6.73 |
| GFASv1.2 | November 1–15, 2022[b] | 14 days | 111 | 10.0 | 5.04 | 0.517 |
| | October 15–November 15, 2022[b] | 1 month | 182 | 16.3 | 8.23 | 0.843 |
| | October 1–December 1, 2022[b] | 2 months | 211 | 18.9 | 9.54 | 0.981 |
| Liu et al. (2020) | Post monsoon, 2003–2018 | 2 months | 791 ± 225 | | 65 ± 18 | 5.6 ± 1.6 |
| Beig et al. (2020) | October–November 2018 | 2 months | | 141.65 | | |

[a]Optimization was not performed from October 15 to November 1, 2022, but the general underestimation value of GFAS (6.99) was applied to boost emissions during the period for comparison with other emission datasets.

[b]Starting from 00:00 UTC until 00:00 UTC

As reported by van der Werf et al. (2017), Liu et al. (2020), and Wiedinmyer et al. (2023), additional use of VIIRS data with finer resolution ($\Delta x$ = 375 m) to capture smaller scale fires, such as CRB, substantially increased emissions over South Asia compared to using only MODIS data ($\Delta x$ = 1 km), such as GFASv1.2 (Kaiser et al., 2012). MODIS-based GFAS was used in emission optimization herein; however, fire emissions should have been underestimated in the presence of

clouds in VIIRS- and MODIS-based inventories such as FINNv2.5 (Wiedinmyer et al., 2023) and the Global Fire Emissions Database version 4s (GFED4s; van der Werf et al., 2017).Therefore, the emission optimization was necessary in any case, and the conclusions reached may have been the same but with different $x_m$ values, even if FINNv2.5 or GFED4s data were used for the emission optimization instead of GFAS. However, since the current emission optimization was based on period-mean values of GFAS emissions (Fig. 2), optimization may be improved by using FINNv2.5 or GFED4s, which may better

represent the spatial distributions of CRB than GFAS data.

## 3.4 Reconstructed air mass movement of anthropogenic and CRB PM$_{2.5}$ in November 2022

The time series of horizontal distributions of simulated surface PM$_{2.5}$ and satellite observations during the Plume 1 and Plume 2 periods are illustrated in Figs. 10 and 11, respectively. The major features were already described in detail and are summarized here. The general features of MODIS AOD and TROPOMI UVAI were similar, with higher resolution and

more available data in the presence of clouds for TROPOMI UVAI. Although direct comparisons between the simulated and TROPOMI observed UVAI were not performed in this study, the plume shapes were similar during the Plume 1 period (Fig. 10). Cyclonic wind fields carried pollutants from the PHD region toward Pakistan on November 1, 2022; the air mass stagnated around the Punjab region on November 2, 2022; northwesterly winds prevailed over the PHD, transporting pollutants from Punjab, to Haryana, to Delhi NCR on November 3, 2022 (Plume 1A); pollutants once carried further



downwind of Delhi blew back on November 4, 2022 (Plume 1B); and cyclonic winds again carried pollutants toward Pakistan on November 5, 2022. These patterns were also observed by AERONET AOD at the Lahore station (Fig. S8a, c, e).

During the Plume 2 period (Fig. 11), the general features of MODIS AOD and TROPOMI UVAI were consistent, although much more data were missing compared to the Plume 1 period (Fig. 10). Even UVAI data were missing during this period. Peaks were observed in surface $PM_{2.5}$ concentrations (Figs. 3, 7, and 8), whereas no peaks were observed over Delhi NCR by the satellites. UVAI was >3 and AOD was >2 over Delhi during the Plume 1 period (Fig. 10), and these values were much lower during the Plume 2 period (Fig. 11). During the Plume 2A event, no simulated influences of CRB were observed over Haryana and Delhi, whereas anthropogenic emissions were high over these regions; however, almost no satellite data were available on November 8, 2022. Likely owing to convective activity, the wind patterns were too complex to be resolved by the low-resolution simulation ($\Delta x = 6$ km), which resulted in the simulated wind fields deviating from those observed. The air was almost stagnant from November 8 to 10, 2022. The shape of plumes between the simulation and TROPOMI were similar on November 9, 2022; however, pollutants from TROPOMI seemed to be more widely distributed than in the simulation. Certainly, the difference in altitude between TROPOMI (column) and the simulation (surface) might have affected horizontal distributions. During the Plume 2B event, northwesterly winds prevailed over the PHD region, and pollutants were transported from Punjab, to Haryana, to Delhi.



**Figure 10:** Spatial distribution of (top to bottom) simulated surface concentrations of anthropogenic PM$_{2.5}$, simulated surface concentrations of PM$_{2.5}$ from Optimized Fire Emission v1.0 (OFEv1.0), simulated surface concentrations of PM$_{2.5}$ from GFAS emission, TROPOMI UV Aerosol Index, and MODIS AOD from November 1 to 5 2022, which includes the Plume 1 period (November 2–4, 2022). The TROPOMI and MODIS maps are identical to those in Fig. 5. The simulation time was 6:00 UTC (11:30 IST), the TROPOMI observation time was 6:30–9:30 UTC (12:00–15:00 IST), and the MODIS time was approximately 10:30 IST.



**Figure 11:** Same as in Fig. 10 but from November 8 to 12 2022, which includes the Plume 2 period (November 8–12, 2022). The TROPOMI and MODIS data are identical to those in Fig. 6.



### 3.5 Contributions of CRB to surface PM$_{2.5}$ concentrations

Based on the simulation using OFEv1.0, the contributions of CRB to surface PM$_{2.5}$ concentrations and the source–receptor relationship among Punjab, Haryana, and Delhi NCR during the two plumes periods in November 2022 are summarized in Fig. 12. The plume duration periods are defined as follows: Plume 1A (November 2, 2022 (12:00 UTC) to

November 3, 2022 (11:00 UTC)), Plume 1B (November 3, 2022 (12:00 UTC) to November 4, 2022 (23:00 UTC)), Plume 2A (November 8, 2022 (00:00 UTC) to November 9, 2022 (23:00 UTC)), and Plume 2B (November 10, 2022 (00:00 UTC) to November 12, 2022 (23:00 UTC)); thus, the durations were 24, 36, 48, and 72 h, respectively. During the Plume 1 period, the observed surface PM$_{2.5}$ concentrations were >50% higher than the half-month average values (November 1–15, 2022 (00:00 UTC)) over the receptor regions (Haryana and Delhi NCR). The mean concentration values and simulated

contributions of CRB emissions from the source regions (Punjab and North Haryana) were higher during the Plume 1A event (274 µg m$^{-3}$ and 50% for Haryana and 419 µg m$^{-3}$ and 50% for Delhi NCR, respectively) than those during the Plume 1B event (219 µg m$^{-3}$ and 40% for Haryana and 324 µg m$^{-3}$ and 50% for Delhi NCR, respectively) because the Plume 1A event was the result of direct transport while the Plume 1B event was the blowback. The mean observed concentration over Punjab and simulated contribution of CRB emission became highest during the Plume 2A event (312 µg m$^{-3}$ and 47%, respectively)

due to the stagnated horizontal wind field predicted by the simulation. Although small increases were observed in surface PM$_{2.5}$ concentrations over receptor regions (Figs. 3c–f, 7c–f, and 8c–f), the optimized emissions could not reproduce the observed high concentration levels, which was likely due to low performance of the meteorological simulation of air flows associated with rainfall and convection. Thus, the source–receptor relationship among the three regions could not be identified for the Plume 2A event in this study. During the Plume 2B event, PM$_{2.5}$ from CRB emissions was directly

transported from the source region to the receptor region, and the contributions become highest in Haryana (56%) and Delhi NCR (55%), whereas the observed PM$_{2.5}$ concentrations were not very large compared to those in the Plume 1 period or the half-month averages. This result may have been due to the larger mean wind speeds during the Plume 2B event (November 11–12, 2022, in Fig. 11) than during the Plume 1A event (November 3, 2022, in Fig. 10) or the amount of CRB emissions in Plume 2B was smaller than that in Plume 1A.



**Figure 12:** Schematic illustration of the contributions of CRB to surface PM$_{2.5}$ concentrations and the source–receptor relationship among Punjab, Haryana, and Delhi NCR during the (left) Plume 1 and (right) Plume 2 periods in November 2022. The period and duration of each plume (1A: November 2, 2022, 12:00 UTC to November 3, 2022, 11:00 UTC, 24 h; 1B: November 3, 2022, 12:00 UTC to November 4, 2022, 23:00 UTC, 36 h; 2A: November 8, 2022, 00:00 UTC to November 9, 2022, 23:00 UTC, 48 h; and 2C: November 10, 2022, 00:00 UTC to November 12, 2022, 23:00 UTC, 72 h) are noted in the figure. Blue circles, orange circles, and arrows indicate roughly simplified CRB emission flux, high concentration plumes, and wind direction, respectively, predicted by the simulation using OFEv1.0. The observed mean PM$_{2.5}$ concentrations are indicated in black or red, and gray values with brackets indicate the observed mean PM$_{2.5}$ for the first half of November 2022 (November 1–15, 2022, 00:00 UTC). The values in red indicate when values exceed the periodical mean values by 50%. The contributions of CRB from areas 1 to 5, namely, NP, WP, CP, EP, and EP&NH, are reported in % for each region and plume. The contributions of fire emissions from outside the PHD region are not included. The period-mean contributions of CRB from areas 1 to 5 were 34%, 33%, and 30% over Punjab, Haryana, and Delhi NCR, respectively.



## 4 Conclusions and future remarks

The impact of post monsoon CRB on surface $PM_{2.5}$ concentrations over the PHD region in North India was investigated using a regional meteorology–chemistry model (NHM-Chem), a high-density in situ surface observation network comprising CUPI-G stations, and the emission optimization technique. Emission optimization was applied for the Plume 1 (November 2–4, 2022) and Plume 2 (November 8–12, 2022) periods identified by Singh et al. (2023) using CUPI-G and meteorological analysis data.

In the source region (Punjab state), almost no enhancements were observed in surface $PM_{2.5}$ concentrations in CPCB stations mostly situated in big cities, whereas substantial increases associated with CRB were observed in CUPI-G data from stations in rural and farmland areas. Employing the CUPI-G data from Punjab enabled us to obtain optimized CRB emissions (Optimized Fire Emission v1.0; OFEv1.0) from November 1 to 15, 2022, which substantially improved the $PM_{2.5}$ simulation over the PHD region compared to using GFAS emission data. Satellite-derived fire emissions are measured in the daytime on daily basis by polar-orbiting satellites; thus, emissions at nighttime have been previously unavailable. However, unlike forest fires, each CRB event lasts only a few hours; hence, information of diurnal variation may be crucial. In fact, some farmers ignite fires after sunset (Liu et al., 2020). Thus, emission optimization was performed at 12 h resolution on a daily basis in the daytime (5:30–17:30 IST, or AM, 00:00–12:00 UTC) and nighttime (17:30–5:30 IST, or PM, 12:00–24:00 UTC).

The major findings of the study are summarized as follows:

- The total CO and $PM_{2.5}$ emissions of OFEv1.0 over the PHD region from November 1 to 15, 2022 were 963 Gg and 86.1 Gg, respectively, which was 8.6 times larger than the original GFAS emissions. OFEv1.0 boosted CRB emissions that were substantially underestimated due to clouds or thick haze on November 8 and 10, 2022. The total emissions of OFEv1.0 were consistent with other relevant inventories for $PM_{2.5}$, OC, and BC, but higher for CO. Optimized daytime and nighttime emissions differed greatly, indicating that consideration of diurnal variations is crucial in emission estimations. Daytime emissions were larger than nighttime emissions on some days but not others, indicating that diurnal variation shape may differ for each day.

- Using OFEv1.0 and NHM-Chem, the half-month (November 1–15, 2022) mean contributions of CRB to the surface $PM_{2.5}$ concentrations over Punjab, Haryana, and Delhi NCR were 34%, 33%, and 30%, respectively.

- The Plume 1 period was divided into two events, including Plumes 1A (November 2–3, 2022) and 1B (November 3–4, 2022). During the Plume 1A event, northwesterly winds prevailed over the PHD region and station-mean concentrations became larger in Haryana (274 µg m$^{-3}$) and Delhi NCR (491 µg m$^{-3}$), and the CRB contributions were




both 50%. The Plume 1B event was smaller than the Plume 1A event because it was a blowback of pollutants once carried further downwind of Delhi. The station-mean concentrations of Haryana and Delhi NCR were 219 and 324 µg m$^{-3}$, respectively, and the CRB contributions were 40% and 43%, respectively.

· Similarly, the Plume 2 period was divided into two events, including Plumes 2A (November 8–9, 2022) and 2B (November 10–12, 2022). The Plume 2B event was similar to the Plume 1A event (direct transport of pollutants from Punjab to Delhi due to northwesterly winds) with the similar CRB contributions (56% in Haryana and 55% in Delhi); however, the PM$_{2.5}$ concentration was not as high as in Plume 1A. During the Plume 2A event, the air mass stagnated around the Punjab region, resulting in a high CRB contribution in Punjab (47%). Although the observed PM$_{2.5}$ in
Haryana was increased in Plume 2A, the simulation did not reproduce the increased emissions, likely due to low performance of the meteorological simulation.

Future issues are itemized as follows:

· The results of this study were obtained using a single model. Multi-model analysis is indispensable for better predictions and quantification of prediction uncertainties using different boundary conditions, meteorological models
with different physical schemes, chemical transport models with different chemical schemes, different emission inventories, and different optimization techniques.

· Post monsoon CRB emissions have negligible impact on surface PM$_{2.5}$ on an annual basis and thus may not cause substantial long-term health effects (Guttikunda et al., 2013; Ghude et al., 2016). However, short-term exposure of
vulnerable populations to high aerosol concentrations may lead to the need for emergency medical care and death (Krishna et al., 2021). Our model simulation using optimized emissions indicated a 50% contribution to short-term exposure on a mass basis. While the 50% value is specific to mass, corresponding information on a toxicity basis has not been determined. Toxicity per unit of aerosol mass may vary substantially depending on the chemical compound, size, mixing state (e.g., Das et al. 2020, 2021, 2023; Ching and Kajino, 2018) and emission source (e.g., Fushimi et al.,
2021; Kajino et al., 2024). Further studies are warranted to determine the associations between chemical compounds and toxicity in order to perform accurate health impact studies.



**Code and data availability**

The NHM-Chem source code is available subject to a licensing agreement with the Japan Meteorological Agency. Further information is available at https://www.mri-jma.go.jp/Dep/glb/nhmchem_model/application_en.html (last accessed: May 7, 2024). The simulation results are freely available. The simulated, observed, and OFEv1.0 data used in the paper are available

at https://doi.org/10.17632/9hs9mtxhh4.1 (Kajino, 2024). The raw observational data can be obtained from the websites, as described in Section 2.

**Author contributions**

The research team consists of following experts: numerical modeling (MK, KI, JC, KY, RI, KK, TK, and PKP), emissions (KI, KY, TS, PKP, and SH), field observations (TS, TK, and YM), satellite analysis (SH), and whole-picture issues in India

(TS, PKP and SH). MK wrote the manuscript, which was reviewed by all coauthors. MK acquired funding for numerical simulations, post processing, and data analysis. MK developed the numerical model and conducted simulations with the assistance of TK. MK conducted the emission optimization under the supervision of KI. MK conducted data analysis and created tables and figures with the assistance of RI. TS, TN, and YM conducted the CUPI-G field measurements with financial support from SH. SH led the Aakash project until March 2023 and PKP has been leading it since April 2023.

**Competing interests**

The authors declare that they have no conflicts of interest.

**Acknowledgments**

This research was mainly supported by the Research Institute for Humanity and Nature (RIHN) (Project No. 14200133: Aakash). It was also supported by a Grant-in-Aid for Scientific Research (KAKENHI) (Grant Numbers JP23H05494,

JP23K17465, and JP19F19402) of the Japan Society for the Promotion of Science, the Environmental Research and Technology Development Fund (Grant Numbers JPMEERF20215003 and JPMEERF20245004) of the Environmental Restoration and Conservation Agency of Japan (ERCA), and the Arctic Challenge for Sustainability II (ArCS II) (Grant Number JPMXD1420318865) from the Ministry of Education, Culture, Sports, Science, and Technology Japan (MEXT). The authors thank Hikaru Araki and Dr. Natsuko Yasutomi of RIHN and Kyoko Kaneba for data collection and processing.



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
