# Peer review of "Impact of post monsoon crop residue burning on PM2.5 over North India: Optimizing emissions using a high-density in situ surface observation network"

_EGUsphere, 2024_

## Author Response (AR1)

Dear Dr. Manish Shrivastava, the handling editor, and two anonymous referees,

We are very grateful for your time on editing and all the constructive comments and deep investigations of two referees. Thanks to their comments, our manuscript has been substantially improved. We reflected all the referees' comments into the revised manuscript. The point-by-point responses (written in blue) to the referees' comments are attached in this letter.

Thank you very much for your time, in advance.

With best regards,
Mizuo Kajino

Dear anonymous referee #1,

We sincerely appreciate your time on review RC2 and your insightful comments on the important points, including careless mistakes and typos. Thanks to your review, our manuscript has been substantially improved, especially for methodology of emission flux optimization. We have carefully addressed all your comments in the revised version.

The region name "Delhi NCR" includes surrounding districts of Delhi. Thus, to be precise, the category name for CPCB and the region name in the maps are changed from "Delhi NCR" to "Delhi", throughout the revised manuscript.

According to referee #2's comment, Figs. 5 and 6 are removed. Figs. 7 to 12 are changed to Figs. 5 to 10, respectively, in the revised manuscript.

Point-by-point responses to your comments are written in blue in this letter.

With best regards,
Mizuo Kajino and Kentaro Ishijima

[1] Authors have used chemical transport model and observations from dense network of low cost sensor to derive emission fluxes at resolution of 12 hours over a region known for crop residue burning (CRB) to find their impact on downwind mega city Delhi. The analysis is very important for the policy makers and air pollution researchers who try to see impact of such activity on air quality, health and cost-benefit analysis. Overall paper is well written, I suggest few minor correction which I believe will make manuscript better and are necessary.

Thank you very much for your evaluation.

[2] Authors use emissions from inventories to calculate non-CRB contribution. One can see that simulation and observation for non-CRB period compare well from Fig. 3. However, authors have used selected stations to prepare those plots. Authors do mention criteria for selection and list of stations selected vis-a-vis not selected in main text and supplementary but I think more details are necessary to replicate their finding by others as well as to understand subjectivity vis-a-vis objectivity in the criteria for

selection of stations. As authors indicate in the absence of these selection, the comparison may not be as good, and in that case CRB fluxes which are interpreted based on difference between observed and simulated concentration will also be different.

Thank you for your comments. There are no objective thresholds of data selection. First, we simply excluded unreliable data, which have sudden gaps or zero drift. For $PM_{2.5}$, most data can be used, more than 70% of station data (78 stations out of 107 stations) are included for the analysis, and there are not many differences between $PM_{2.5}$ for selected sites and all sites in both CPCB and CUPI-G (please compare left and right panels in Fig. S2). Thus, we presume that the optimized emissions may not be so much different either using all stations or using only selected stations. However, because the purpose of our study is to estimate the CRB emission flux, we were especially careful on the selection of data during the non CRB period, October 15-28. If there are big gaps in the simulated and observed data during the non CRB period, our optimization will be misled to fill the gap by altering CRB emissions, which may not be realistic. We inserted the following statement in the 5th paragraph of Sect. 2.3 as:

 "In other words, if there are big gaps in the simulated and observed data during the non CRB period, our optimization would be misled to fill the gaps by altering CRB emissions."

Also, I visited not all but some of CPCB stations in Delhi, where wind speed data are very low, or solar radiation data are always much lower than the clear sky conditions. Most stations are well situated in open places, but some are situated in locations surrounded by tall buildings. Those stations are suitable for human exposure studies, but not for the NHM-Chem study, because our model does not simulate air quality inside the urban canopy. This is also the reason why we excluded such stations for the emission optimization.

Still, it is not our motivation to show which stations are suitable or not suitable for the emission optimization, we didn't show time series of meteorological and air quality data of individual stations but only showed the list of stations we used as in Table S1.
re
Thank you very much for your understanding.

[3] Emission flux estimates in top down approach are highly sensitive to simulation of vertical distribution of species which in turn is highly sensitive to simulation of boundary layer dynamics. Authors should show either using previously published studies for their model or from this study, how good were boundary layer simulation over South Asia and preferably over North India.

Thank you for your suggestion. As shown in Fig. S9, we compared the simulated vertical profiles of $CO$ and $O_3$ against observation data made by IAGOS, the commercial aircraft monitoring campaign. Based on the comparison, we made a new section 3.6 to discuss uncertainty in top-down emission flux due to boundary layer simulation. Please see Sect. 3.6, "Uncertainties in our top-down emission optimization approach".

[4] Top down approach for flux estimate is also sensitive error in observation. Why authors have not mentioned errors in observations explicitly, they imply temporal standard deviation of 3 hours exceeds error in observation. This may be true for random error but if the observation had systematic error then it will still affect the calculation of emission flux. Authors should provide information on error in observation and discussion on how they would affect emission flux estimate.

Honestly speaking, this is my first paper to conduct emission optimization, and so I did not realize showing observation errors are critically important.

This observational data uncertainty generally consists of observational errors and spatial representativeness errors. In the analysis of the global to subcontinental scale carbon dioxide flux (Maki et al., 2010), observational errors are clearly smaller compared to spatial representativeness errors. Therefore, they mainly estimated spatial representativeness errors for each observation site based on the difference from the fitting curve of the monthly average carbon dioxide observations. In the analysis of dust emissions in the Gobi Desert (Maki et al., 2011), they used $PM_{10}$ observations over a wide range from China to Japan (considering observations above a certain value as dust). Although they assumed that there were no significant differences in instrument errors spatially, they considered spatial representativeness errors to vary by location and set the observational data uncertainty to 10% of the observed $PM_{10}$ concentrations. This paper, since we are using an observation network in a relatively small area in northern India, we believe it is reasonable to treat observation uncertainty (spatial representativeness errors and observational errors) uniformly.

We elaborated on this in the 4th paragraph of Sect. 2.3 by inviting an expert of inverse

modeling Dr. Takashi Maki to the coauthor. Uncertainty in emission flux estimates are discussed in Sect. 3.6.

References:

Maki, T., Ikegami, M., Fujita, T., Hirahara, T., Yamada, K., Mori, K., Takeuchi, A., Tsutsumi, Y., Suda, K. and Conway, T. J.: New technique to analyse global distributions of CO2 concentrations and fluxes from non-processed observation data, Tellus, 62B, 797-809, doi:10.1111/j.1600-0889.2010.00488.x, 2010.

Maki, T., Tanaka, T. Y., Sekiyama, T. T. and Mikami, M.: The impact of ground-based observations on the inverse technique of aeolian dust aerosol, SOLA, 7A, 21-24, doi:10.2151/sola.7A-006, 2011.

[5] The discussion regarding emission optimization is not very clear -- at least to me. The point-by-point responses are written as below from [5a] to [5c]. We also had similar comments by Referee 2. Please refer to the RC1 comments #6 together with our corresponding replies.

[5a] Based on the discussion that follows in and after section 2.3, $x_m$ simply appears to be ratio of observed concentration to simulated concentration or a multiplier when multiplied to simulated concentration excluding non-CRB concentration, it matches simulated concentration to observed concentration. I might be wrong to interpret $x_m$ in this manner but If I am correct about $x_m$ as ratio, then authors should explain

$x_m$ is the scaling parameter for each CRB emission flux tag. $S_m$ and $S_0$ are the $PM_{2.5}$ simulation data with and without the CRB emission tags, so that $S_m$ minus $S_0$ is considered to be $PM_{2.5}$ from CRB emission tag (this is so-called the brute-force concept with a zeroed-out test to derive source contributions, as explained in our reply to your Comment #7).

[5b] Are optimized emission fluxes are simply scaled up emission fluxes based on that ratio?

That's why, yes.

[5c] How would second term in the equation 1 would imply importance of apriori in the cost function?

This can be altered by setting the upper and lower limits ($u_m$, $l_m$). As the limits are broader, the relative importance of the second term is smaller so that the algorithm

tried to minimize deviations in the first term. Anyway, relative importance of the first and second terms in the cost function depends on the relative magnitudes of $\sigma_n$ and $(u_m, l_m)$. The sensitivities of those parameters to emission optimization were tested by altering $(u_m, l_m)$ as (0.5, 2), (0.1, 10), and (0.01, 100), as presented in the new section, Sect. 3.6.

[6] Authors write that the denominator in the second term of Eq 1 ($u_m$, $l_m$) were taken as (2.0 and 0.5) and in an multistep process calculations were repeated until $x_m$ values were between 0.009 and 1.001. Later-on in the result section authors show values of $x_m$ ranging from 0 to 69 with several of them more than 1. The text is not clear enough to describe how the values of $x_m$ get these high values. Authors should clarify $x_m$ calculations and how they can have such high value in spite of their set criteria.

$x_m$ was obtained after multiple steps of optimization. Suppose $x_m$ obtained after the *i*-th step is defined as $x_{m,i}$, optimization was repeated until *k*-th step when $x_{m,n}$ get enough close to 1, and $x_m = x_{m,1} \, x_{m,2} \, x_{m,3} \ldots x_{m,k}$. When $x_{m,i}$ (*i* = 1, 2, … ) are (2.0, 2.0, 2.0, …, 1.5, 1.05, 1.005, 1.001, 1.0001, 1.0001, …), $x_m$ larger than 2.0 is obtained.

We inserted the following description in the 3rd paragraph of Sect. 2.3:

"Let xm,i be the value of xm obtained in the i-th step. This multi-step optimization was repeated until 0.009 < xm,i < 1.001 was obtained for all m valuesat the k-th step, so that the final xm was obtained as:

$$x_m = \prod_{i=1}^{k} x_{m,i}. \tag{2}"$$

[7] Page 19 lines 5-10: The description of calculation of anthropogenic contribution is not clear.  Why the 20% emission reduction, why not 25% or 40%? Which emissions -- All the emissions or the emission that are newly calculated over and above non emissions? What does multiply by 5 implies? Is the simulated concentrations multiplied by 5? Why would difference between control and reduced/multiplied concentrations would imply anthropogenic contribution?

This is so-called a brute-force method to estimate source-receptor relationship. If zeroed-out sensitivity test (anthropogenic $NO_x$, $SO_x$, $NH_x$, and VOC concentrations are set all zero) was conducted, the oxidant fields such as $O_3$ and OH to produce secondary components of $PM_{2.5}$ (such as $SO_4^{2-}$, $NH_4^+$, and $SO_4^{2-}$) are significantly altered (i.e.,

significantly underestimated). Thus 20% reduction run was conducted, so as not to change the oxidant fields much, and 5 times the deviations between the control run and the 20% reduction run are defined as the source contributions (anthropogenic contributions) (20% times 5 = 100%). For the sensitivity runs of biomass burning sources, zeroed-out runs were conducted because it may not alter the oxidant fields significantly, because the contributions from other sources such as anthropogenic sources may be more to the oxidant fields.

Yes, in fact, the brute-force method with a 25% reduction run then 4 times the deviation (Lin et al., 2008) or a 10% reduction then 10 times the deviation (Bartinicki, 1999), existed. As the reduction rate is smaller, the artifact due to altering oxidant fields get smaller, but the numerical errors become larger (for example, the deviation between 1% reduction run and control run is so small that the S/N ratio is small. Then 100 times the deviation may contain larger noise in their source-receptor estimation). Thus, we reached a consensus that 20% is a suitable value, after our extensive activities for the source-receptor relationship studies in East Asia conducted by the Long-range Transboundary Air Pollutants in Northeast Asia (LTP project) among China, Korea, and Japan (e.g., NIER, 2010, Kajino et al., 2013).

We modified the description of the brute-force method ($2^{nd}$ paragraph of Sect. 3.2) as follows:

"The anthropogenic contribution was calculated using the brute-force method, derived from the difference between the control run and the 20% emission reduction run multiplied by 5 to reduce the effect of nonlinearity in chemical reactions (Kajino et al., 2013): The brute-force method with a zeroed-out simulation (no anthropogenic $NO_x$ and NMVOC) affects oxidant concentrations substantially and thus chemical production rates of secondary $PM_{2.5}$, which deteriorates the calculation of source contribution estimations."

Refs.:

Bartinicki, J., 1999. Computing Source-receptor Matrices with the EMEP Eulerian Acid Deposition Model. EMEP/MSC-W, Note 5/99. Norwegian Meteorological Institute, Oslo, Norway, p. 37.
Kajino, M., Sato, K., Inomata, Y. and Ueda, H., 2013. Source-receptor relationships of nitrate in Norheast Asia and influence of sea salt on the long-range transport of

nitrate, Atmos. Environ., 79, 67-78, doi:10.1016/j.atmosenv.2013.06.024.

Lin, M., Oki, T., Bengtsson, M., Kanae, S., Holloway, T., Streets, D.G., 2008. Long-range transport of acidifying substances in East Asia — Part II source-receptor relationships. Atmospheric Environment 42, 5956-5967.

National Institute of Environmental Research (NIER), 2010. The 10th Year's Joint Research on Long-range Transboundary Air Pollutants in Northeast Asia. Annual Report of LTP Project 2009. NIER, Korea.

**Minor Issues:**

[8] Throughout the text and in abstract and in conclusion authors mention their model as NHM-Chem. However, in the methodology section, authors mention that only the Chem part is used from NHM-Chem whereas meteorology is simulated using WRF. Since, the WRF model is also available with its own Chem version known as WRF-Chem. It is imperative that authors should briefly describe why instead of WRF's Chem they chose the different Chem model and instead of NHM's meteorology, they chose WRF's meteorology. Also they need to come-up with better nomanclature then NHM-Chem since WRF-Chem would be misunderstood as default WRF-Chem model and NHM-Chem would be misunderstood as default NHM meteorology.

I agree with you. The offline coupling between WRF and the CTM part of NHM-Chem was referred to as NHM(WRF)-Chem, as defined in the 1st paragraph of Sect. 2.1. The reason why we selected NHM(WRF)-Chem is because it was performed slightly better than NHM-Chem, in terms of surface $PM_{2.5}$. The reason why we used NHM-Chem and did not use WRF-Chem is because NHM-Chem is our model. The reason why we selected NHM(WRF)-Chem was explained in the last sentence of 1st paragraph of Sect. 2.1 as "because NHM(WRF)-Chem exhibited a slightly superior performance compared to NHM-Chem when evaluated against observed time series of surface $PM_{2.5}$ in North India" We modified the nomenclature elsewhere in the entire manuscript.

[9] Page 3 Line 29: Authors write "polar orbiting satellites travel once during the day ..". Most polar orbiting satellites including Terra and Aqua travel twice over a place in 24 hours, once during daytime and once during nighttime. Satellites use thermal channels to find fire hot-spot which in principle should work better in night. Should authors clarify this point in the text and explain why there are no nighttime data for the fire hotspots?

Thank you for the correction. The statement was totally wrong. Burned area (BA) product is obtained only in the daytime, but the polar orbiting satellites travel twice per day. The statement was based on my presumption that the biomass burning emissions

are primarily derived using BA (such as GFED and FINN), but they are also derived by other parameters such as fire radiative power (FRP), that are efficiently measured during the night. In fact, GFAS emission we mainly used in this study was based FRP and their daily emission is the combination of daytime and nighttime emissions (Kaiser et al., 2012). Thus, the sentence was changed to "polar-orbiting satellites travel twice during the day and night, usually around noon and midnight", in the 3rd paragraph of Sect. 1. The similar statement in the same paragraph, "Nevertheless, nighttime observational data are unavailable even though some farmers ignite fires after sunset (Liu et al., 2020)" was deleted. Similarly, there was another sentence in the 2nd paragraph of Sect. 4: "Satellite-derived fire emissions are measured in the daytime on daily basis by polar-orbiting satellites; thus, emissions at nighttime have been previously unavailable.", which was changed to "Diurnal variations of satellite-derived fire emissions have been previously unavailable".

[10] Page 9 Eq 1: There is an extra plus sign in the equation. Also, the symbol $\sigma_0$ should be $\sigma_n$ if separate sigma was used for each observational data. Or otherwise mention how the $\sigma_0$ is calculated.

Thank you for finding the error. I deleted the extra plus sign. As for $\sigma_0$, it is not sigma_zero, but sigma_O (O of Observation). Anyway, since it was misleading, I changed it from sigma_zero to sigma_n. Relating to your comment #4, how sigma_O was calculated was explained in the 4th paragraph of Sect. 2.3.

[11] Fig 7e: GFAS is written as GFSS
Thank you for finding the typo. I modified it.

Dear anonymous referee #2,

We are very grateful for your time on review RC1 and your comments on the important points. Thanks to your review, our manuscript has been substantially improved, especially for presentations and readability. We have carefully addressed all your comments in the revised version.

The region name "Delhi NCR" includes surrounding districts of Delhi. Thus, to be precise, the category name for CPCB and the region name in the maps are changed from "Delhi NCR" to "Delhi", throughout the revised manuscript.

Point-by-point responses to your comments are written in blue in this letter.

With best regards,
Mizuo Kajino and Kentaro Ishijima

**General Remarks:**

[1] This study develops an optimized emission estimate of crop residue burning in North India in October and November 2022. This study develops this emissions inventory by using a top-down approach, which according to the authors, has not been done for post monsoon crop residue burning in North India. I find this to be an important contribution to the understanding of crop residue burning emissions and the impact of these emissions on PM$_{2.5}$ in North India. However, I believe the results could be made more clear and concise. In the present state, the scientific significance of the paper is inhibited by the quality of the discussion and presentation. I recommend major revisions.

Thank you for your evaluation. We revised our manuscript following your comments.

**Specific Comments:**

*Introduction*

[2] I find the introduction to be a bit too brief. There could be more discussion on the estimates of PM from CRB in prior studies and reasons for why a consensus has yet to been reached. Additionally, it would be helpful to summarize the emissions estimates for global biomass burning inventories (eg. GFAS, GFED, QFED, FINN) during the post monsoon period in this region as this would help to provide additional context for the

inventories used in this study (both GFAS and the developed top-down CRB inventory).

Thank you very much. It is well summarized by Wiedinmyer et al. (2023) so a part of their findings was incorporated in 1. Introduction as follows:

"Wiedinmyer et al. (2023) summarized commonly used emission inventories of open biomass burning such as Fire Inventory from National Center for Atmospheric Research (FINNv2.5; Wiedenmier et al., 2023), Global Fire Emissions Database (GFED4.0s; van der Werf et al., 2017), Fire Energetics and Emissions Research (FEER; Ichoku and Ellison, 2014), the Global Fire Assimilation System (GFASv1.2; Kaiser et al., 2012), and the Quick Fire Emissions Dataset (QFED v2.5; Darmenov and da Silva, 2015). They reported that there are substantial variations in the estimation of open biomass burning emissions among them over the South and Southeast Asian region and the relative magnitudes also varied substantially among species (See. Fig. 4 of Windinmyer et al., 2023). They concluded that determining the cause of different fire emissions in the region is a target for their future research."

[3] I also think it would be beneficial to further discuss specific advantages of using the top-down approach over the bottom-up approach in terms of estimating biomass burning emissions, and to the extent prior studies allow in the context of CRB or agricultural burning in other parts of the world.

Thank you for your suggestion. We inserted the following statements in 4. Conclusion and future remarks:

"Similar issues, namely, "satellite-based observations are underestimated by clouds or thick smoke/haze" and "ground-based observations are mostly in cities and not near the fire emission source regions", may also exist in other parts of the world. Our methodology, top-down estimation of emissions using distributed low-cost sensors, can be applied to other cases in the world."

[4] I think expanding the final paragraph of the introduction to layout the format of the rest of the paper would improve the ability to follow the results. This could be accomplished through stating what general topics are discussed in each section of the results.

Thank you for your suggestion. General structure of the manuscript is additionally described in the final paragraph of Sect. 1. Introduction, as follows:

"The model, observation data, and the emission optimization method are described in Sect. 2.1, 2.2, and 2.3, respectively. The model validation using the observation data is presented in Sect. 3.1. The results of emission optimization are intercompared with each other and the best estimate is referred to as Optimized Fire Emission v1.0 (OFEv1.0) in Sect. 3.2. The OFEv1.0 data is compared against other bottom-up inventories in Sect. 3.3. The reconstructed movements of polluted air masses are presented in Sect. 3.4 and the contributions of CRB to surface PM2.5 concentrations are quantified in Sect. 3.5. Uncertainties in OFEv1.0 associated with parameters used in the optimization method and with the planetary boundary layer (PBL) simulations are discussed in Sect. 3.6. Concluding remarks and future issues are summarized in Sect. 4."

*Methods*

[5] Figure 1 could be improved. The color bar for the terrain could be the same between the left and right panels. The coloring of the observation sites could be more informative for the interpretation of later figures by using different shapes to indicate CPCB and CUPI-G sites and coloring to indicate Source, Intermediate, and Delhi NCR sites.

We modified Fig. 1 accordingly as well as the captions: using the same coloring in the terrain of both domains, using colored circles with regional categories for CUPI-G, and using squares for CPCB. We also modified Fig. S1 in the same manner.

[6] Aspects of the emission optimization using tagged simulations need to be clarified or explained further.

We improved the sentences accordingly. The point-by-point responses are written as below from [6a] to [6e]. We also had similar comments by Referee 1. Please refer to the RC2 comments #4, #5, and #6 together with our corresponding replies.

[6a] Equation 1 needs to be better explained in terms of what the variables are in this specific study. For example, O and S are observed and simulated data of CO, or $PM_{2.5}$.

Changed to "where O and S are the observed and simulated $PM_{2.5}$ surface concentrations" in the first paragraph of Sect. 2.3.

[6b] The description of the method of optimization to minimize the cost function is too vague.

We inserted additional descriptions in the 1$^{st}$, 3$^{rd}$, 4$^{th}$, and 5$^{th}$ paragraphs of Sects. 2.3.

We hope it is now enough compelling.

[6c] Is the No_CRB simulation simply a simulation with no GFAS emissions? This needs to be clear.

Yes, it is. "(i.e., without GFAS emissions)" are inserted in the second paragraph of Sect. 2.3.

[6d] Was a threshold of $PM_{2.5}$ bias during the period not affected by CRB used to exclude observation sites? If so what was this threshold and how was it selected?

No, there are no such thresholds of data selection. We simply excluded unreliable data, which have sudden gaps or zero drift. For $PM_{2.5}$, most data can be used, more than 70% of station data (78 stations out of 107 stations) are included for the analysis, and there are not many differences between $PM_{2.5}$ for selected sites and all sites in both CPCB and CUPI-G (please compare left and right panels in Fig. S2). Because the purpose of our study is to estimate the CRB emission flux, we were especially careful on the selection of data during the non CRB period, October 15-28. If there are big gaps in the simulated and observed data during the non CRB period, our optimization will be misled to fill the gap by altering CRB emissions, which may not be realistic. We inserted the following statement in the 5th paragraph of Sect. 2.3 as:

"In other words, if there are big gaps in the simulated and observed data during the non CRB period, our optimization would be misled to fill the gaps by altering CRB emissions."

[6e] Was the discussion on spatial temporal representativeness of an observation used to exclude sites?

Yes, I visited not all but some of CPCB sites in Delhi, where wind speed data are very low (almost zero for longer period), or solar radiation data are always much lower than the clear sky conditions. Most stations are well situated in open places, but some are situated in locations surrounded by tall buildings. Those stations are suitable for human exposure studies, but not for the NHM-Chem study, because our model does not simulate air quality inside the urban canopy.

*Results*
[7] I believe the results could be restructured to better highlight the impact of the optimized emission estimates developed in this study. Also, ensure that simulation

names are consistent with Table 1 and throughout the results section.

Thank you for your advice. The point-by-point responses are written as below from [7a] to [7i]. We also confirmed that the simulation names in Table 1 are now consistent throughout the manuscript.

[7a] Section 3.1 Time series comparisons: This section would better be entitled "Whole Period GFAS Simulation Evaluation", or something similar. However, I find this section to be too long. Figures 3 is better discussed in the context of the impact of the optimized section (Figures 7 and 8 show the GFAS simulation). While the point of Figure 5 and 6 to explain reasons for potential biases in the GFAS inventory during the plume events is valuable, it could be an SI figure that is discussed with the results of Figure 9.

Thank you for your advice. This section is also to evaluate meteorological simulation, so that title was changed to "Evaluation of meteorological and chemical simulations with GFAS". We admitted that the section was too long and redundant, and so we removed Figs. 5 and 6 as they are the duplications of Figs. S4 to S7. Accordingly, Figs. 7 to 12 were changed to Figs. 5 to 10.

[7b] Table 2: The discussion states that $x_m$ values were obtained for different temporal tags, could these be presented in the SI?

We made Table S2 to show all $x_m$ values temporally and spatially. The following sentence was inserted to the 2nd paragraph of Sect. 3.2:

"Optimized $x_m$ values for all temporal and areal tags are presented in Table S2."

[7c] I think the discussion in Section 3.2 could be improved by having Figure 7 be altered to have 3 panels: Source Region $PM_{2.5}$, Intermediate Region $PM_{2.5}$, Delhi NCR Region $PM_{2.5}$. On each panel would be a line for observation, GFAS run, CPCB+CUPI-G, CPCB+CUPI-G+CUPI-G Punjab and CPCB+CUPI-G+CUPI-G P89.

We modified it accordingly. We found it much better than the original structure. Thank you for your advice. The new figure number is now Fig. 5.

[7d] Figure 8 could then be changed to show the simulated contribution of different emission sources in each of the four simulations.

We modified it accordingly. We found it much better than the original structure. Thank you again for your advice. The new figure number is now Fig. 6.

[7e] There needs to be better discussion and quantification of the PM$_{2.5}$ biases. What is meant by the optimized run substantially improved biases discussed in the first paragraph of page 21. Table 3 could be introduced here.

Thank you for your suggestion. The phrase "Statistical scores between the simulated and observed surface concentrations are listed in Table 3." as moved to the relevant place (now 3$^{rd}$ paragraph of Sect. 3.2).

[7f] Table 3 could be improved by including mean biases of PM$_{2.5}$.

Thank you for your suggestion. We added RMSE in Table 3 as one of the indicators to evaluate deviations between the simulation and observation. We showed RMSE instead of MB, as MB can be small if there are large positive and small negative values. RMSE does not tell if there are positive or negative biases, but Sim./Obs. ratio can imply over- or under-prediction. Thanks to you for your understanding.

[7g] The discussion of the contribution of different tagged source regions on page 23 could benefit from more quantitative discussion rather than vague substantially increased or became higher language currently used.

We inserted numbers into 5 sentences in the 7$^{th}$ paragraph (2$^{nd}$ last paragraph) of Sect. 3.2, namely:

"certain contribution of CRB emissions was continuously observed in the "GFAS" simulation in the source region (10-20%,",
"affected during the two plume events in the intermediate region and Delhi NCR by 10–20%",
"contributions from SP&NH were substantially increased up to approximately 50%",
"whereas contributions from EP were also increased up to approximately 60%",
And "the contributions from CP were enhanced approximately from 20% to 50%".

[7h] Table 4 would benefit from including FINN or GFED data even though it was not used for the optimization.

We inserted the values from FINNv2.5 and GFED4s.

[7i] Section 3.4: would it be possible to plot simulated AOD in the first three rows of Figures 10 and 11 instead of surface PM$_{2.5}$?

It would be a nicer idea to compare simulated and observed AOD. However, there is

still a problem with our model that the simulated light extinction to mass (or volume) ratios are generally underestimated so as AOD (please refer to Fig. S8 or Table 4 of our model description paper, Kajino et al., 2019). Currently we are working to improve the AOD simulation, which is validated in the future study. Also, the main focus of this study is the prediction of not AOD but surface $PM_{2.5}$, and satellite AOD (and UVAI) are used just as a reference. Thank you for your understanding.

Kajino, M., Deushi, M., Sekiyama, T. T., Oshima, N., Yumimoto, K., Tanaka, T. Y., Ching, J., Hashimoto, A., Yamamoto, T., Ikegami, M., Kamada, A., Miyashita, M., Inomata, Y., Shima, S.-I., Takami, A., Shimizu, A. and Hatakeyama, S.: NHM-Chem, the Japan meteorological agency's regional meteorology – chemistry model: Model evaluations toward the consistent predictions of the chemical, physical, and optical properties of aerosols, J. Meteorol. Soc. Japan, 97(2), doi:10.2151/JMSJ.2019-020, 2019.

**Technical Corrections:**

*Abstract*

[8] Line 8: accounting for 80% of the CRB emissions

We changed the sentence accordingly.

[9] Line 9: "OFEv1.0 substantially boosted emissions," relative to what? GFAS?

Yes, it is GFAS. We modified the whole sentence as follows: "OFEv1.0 substantially increased emissions relative to those calculated from satellite fire observation data (prior emissions)".

[10] Line 16-17: Last two sentences of the abstract seem unnecessary, could be moved or re-worded.

We followed your suggestion by removing the two sentences, as it seems too natural to state: "Don't believe simulation results fully as it is based on a single model and a single emission inventory."

*Introduction*

[11] Page 3, Line 8: Delhi's air quality worsens during the post monsoon to winter period because of weaker wind speeds than other times of the year (Citation?), increased emissions from space heaters (Guttikunda and Gurjar, 2012; Chowdhury et al., 2017, 2019; Guttikunda et al., 2023), use of fireworks during Diwali festivities (Singh et al.,

2019), and crop residue burning (CRB) upwind of Delhi (CITATION).

Guttikunda and Gurjar (2012), Chowdhury (2017), Guttikunda (2023) stated emissions from space heaters but also meteorology, so that no additional citation for the first sentence. According to ", and crop residue burning (CRB) upwind of Delhi (CITATION)", we modified the entire sentences as follows:

"Delhi's air quality worsens during the post monsoon to winter period, which is associated with (1) weaker wind speeds than other times of the year and increased emissions from space heaters (Guttikunda and Gurjar, 2012; Chowdhury et al., 2017, 2019; Guttikunda et al., 2023), (2) use of fireworks during Diwali festivities (Singh et al., 2019), and (3) crop residue burning (CRB) upwind of Delhi (Cusworth et al., 2018; Beig et al., 2020; Takigawa et al., 2020; Liu et al., 2020; Singh et al., 2023; Mangaraj et al., 2025)."

[12] Page 3, Line 20: list other uncertainties in source apportionment. Is there a citation to support that the largest uncertainty is in the biomass burning inventories?

No. We rephrased the relevant sentence as follows: "Accuracy in emission inventories is crucial for the better prediction of source apportionment using 3D chemical transport models."

[13] Page 4, Line 2: "which resulted in a 109% increase compared to MODIS" Increase in what? CO emissions? $PM_{2.5}$ emissions?

109% increase of Fire Radiative Power. We modify the sentence accordingly.

*Methods*
[14] Page 5, Line 6-Line10: "The CTM part of offline-coupled NHM-Chem can be employed…" This sentence seems to describe possible couplings not used in this study; it could be deleted.

We deleted the sentences accordingly, except offline coupled with WRF, which is used in this study.

[15] Page 7, Line 10: "Open biomass burning emissions were assumed to be constant over time." Does this mean over the entire simulation, or that there was assumed to be no diurnal cycle?

The latter is correct. The relevant sentence was modified to "No diurnal variations are considered in the open biomass burning emissions (GFAS), and their vertical …".

[16] Page 9, Line 21: To optimize CRB emission fluxes, we used…

Modified accordingly.

[17] Page 10, Line 10: The list of sensitivity simulations performed herein are summarized in Table 1.

Corrected.

*Results*
[18] Page 12, Line 16: Fig. 3 presents $PM_{2.5}$ not emissions

Corrected.

[19] Figure 3: Legend entry should say GFAS to avoid confusion with other simulations presented in the study, y-axes labels should say what variable is plotted

The legend "Simulation" was changed to "GFAS run" to be consistent with Fig. 7 (now Fig. 5). Y-axes labels are modified accordingly.

[20] Figure 4: y-axes labels should say what variable is plotted not just the units

Y-axes labels are modified accordingly. Same for Figs. 7 (now 5), 8 (now 6), S2, S3, and S8.

[21] Figure 7e: GFSS run should say GFAS run

Corrected.

[22] Figure 7 caption: (g-l) is not the same as (a-f), say it's the fractional contribution of different emission sources to $PM_{2.5}$

Thank you for your comment. Figs. 7 and 8 (now Figs. 5 and 6) are totally reorganized according to your Comment #7c, so that the legend is now fine.

[23] Figure 7/8 axes labels: the y-axes labels should be informative as to which variable is plotted not just the units of the variable

Corrected. (Figs. 5 and 6, now)

[24] Page 23, Line 6: "even though its magnitude was small," this sentence could be reworded to be clearer

The sentence was changed from

"even though its magnitude was small, the contribution of CRB emissions was continuously observed"

to

"small but certain contribution of CRB emissions was continuously observed".

[25] Page 26, Line 17: Reword the sentence, maybe "maps" is more informative than "horizontal distribution"

Modified accordingly.